

# The Multi-Assumption Architecture and Testbed (MAAT v1.0): Code for ensembles with dynamic model structure including a unified model of leaf-scale C3 photosynthesis

Anthony P. Walker[1], Ming Ye[2], Dan Lu[1], Martin G. De Kauwe[3], Lianhong Gu[1], Belinda E. Medlyn[4], Alistair Rogers[5], and Shawn P. Serbin[5]

[1]Environmental Sciences Division and Climate Change Science Institute, Oak Ridge National Laboratory, Oak Ridge, Tennessee, USA
[2]Department of Earth, Ocean, and Atmospheric Science, Florida State University, Tallahassee, Florida, USA
[3]ARC Centre of Excellence for Climate Extremes, Climate Change Research Centre, University of New South Wales, Sydney, New South Wales, Australia
[4]Hawkesbury Institute for the Environment, Western Sydney University. Locked Bag 1797 Penrith, New South Wales, Australia
[5]Environmental and Climate Sciences Department, Brookhaven National Laboratory, Upton, New York, USA

*Correspondence to:* Anthony P. Walker (walkerap@ornl.gov)

**Abstract.** Computer models are ubiquitous tools used to represent systems across many scientific and engineering domains. For any given system, many computer models exist, each built on different assumptions and demonstrating variability in the ways in which these systems can be represented. This variability is known as epistemic uncertainty, i.e. uncertainty in our knowledge of how these systems operate. Two primary sources of epistemic uncertainty are: 1) uncertain parameter values, and 2) uncertain mathematical representations of the processes that comprise the system. Many formal methods exist to analyse

parameter-based epistemic uncertainty, while process-representation based epistemic uncertainty is often analysed informally and incompletely, or is ignored. In this model description paper we present the Multi-Assumption Architecture and Testbed (MAAT v1.0) designed to formally and more completely analyse process-representation based epistemic uncertainty. MAAT is a modular modelling code that can simply and efficiently vary model structure (process representation) during runtime, allowing the generation of large ensembles that vary in process representation, as well as in parameters, parameter values, and

environmental conditions. MAAT v1.0 approaches epistemic uncertainty through sensitivity anlaysis, assigning variability in model output to processes (process representation and parameters) or to individual parameters. In this model description paper we describe MAAT and by using a simple groundwater model example, verify that the sensitivity analysis algorithms have been correctly implemented. The main system model currently coded in MAAT is a unified, leaf-scale enzyme kinetic model of C3 photosynthesis. We describe the photosynthesis model and the unification of multiple representations of photosynthetic

processes. The numerical solution to leaf-scale photosynthesis is verified and examples of process variability in temperature response functions are provided. For rapid application to new systems, the MAAT algorithms for efficient variation of model structure and sensitivity analysis are agnostic of the specifc system model employed. Therefore MAAT provides a tool for development of novel or 'toy' models in many domains, i.e. not only photosynthesis, facilitating rapid informal and formal

comparison of alternative modelling approaches.



# 1 Introduction

Systems are composed of multiple interacting components and processes, and can exhibit complex and unforseen behaviour. Mathematical computer models are a valuable tool in the study of systems behavior, providing a quantitative approximation of the main features and processes of a system. Computer models are used widely across many scientific and industrial domains,

for example to: explore hypotheses on ecosystem processes (e.g. Comins and McMurtrie, 1993), identify the biophysical factors controlling biological activity (e.g. Walker et al., 2017a), interpolate sparse observations (e.g. Compo et al., 2011), project responses of the Earth System to anthropogenic activity (e.g. Friedlingstein et al., 2014), predict aerodynamic flow over new wing designs (e.g. Jameson et al., 1998), and forecast the weather (e.g. Molteni et al., 1996). Real-world processes (often how two or more variables are related) are included in models using mathematical representations of mechanistic hypotheses or

conceptual, simplifying, or empirical assumptions (see Table 1 for our definition of terms). When multiple plausible assumptions exist for a particular process, a model developer is faced with the choice of which assumption to use in their model (Fig. 1). For a single process, the consequences of this choice can be assessed in a relatively simple way. However, when multiple assumptions exist for multiple processes (e.g. Fig. 1) the options combine in factorial to generate a large number of plausible system models. This large number of plausible system models characterises process representation uncertainty and poses a

challenge to understanding and interpreting predictions for the modelled systems (e.g., Medlyn et al., 2015; Friedlingstein et al., 2014; Beven, 2006).

Process representation uncertainty, a component of epistemic uncertainty (Beven, 2016), is often referred to as model structural uncertainty (e.g., Gupta et al., 2012; Beven, 2016), or conceptual model uncertainty (e.g., Rojas et al., 2008; Dai et al., 2017). While model structural uncertainty is a broadly encompassing term (see Gupta et al., 2012, for an in depth discussion

of the multiple facets of model structural uncertainty), in this paper we use the term process representation uncertainty as it implies hypotheses and assumptions and therefore connects more directly with the language of experiment and observation. Often process representation uncertainty is assessed by analysing the cross-model variability in the ensembles of model inter-comparison projects (Refsgaard et al., 2007; Friedlingstein et al., 2014; Herger et al., 2018). These ensembles can be thought of as ensembles of opportunity, rather than formal assessments of process representation uncertainty. Very often the models in the

ensemble are not independent of one another and a full ensemble of all possible models is never posed. Moreover, reduction of uncertainty requires that researchers identify the processes responsible for cross-model variability in MIPs, which is challenging and time consuming (e.g. see De Kauwe et al., 2013; Medlyn et al., 2015). Incomplete or out-of-date model documentation, modeller specific code, incomplete information for how a particular simulation has been executed, and superficial knowledge of how a model works all contribute to the difficulty of process level analysis in MIPs. A primary reason for this failure is that

adequate tools for the assessment of model sensitivty to variability in process representation are not available.

The multi-assumption architecture and testbed (MAAT v1.0) is a modelling framework that can formally, systematically, and rigorously analyse variability in system model output caused by variability in process-representation. MAAT allows users to specify multiple process representations for multiple processes and can configure the ensemble of all possible combinations of these choices during runtime. The main component of MAAT is a software wrapper and an interface that passes assumptions to



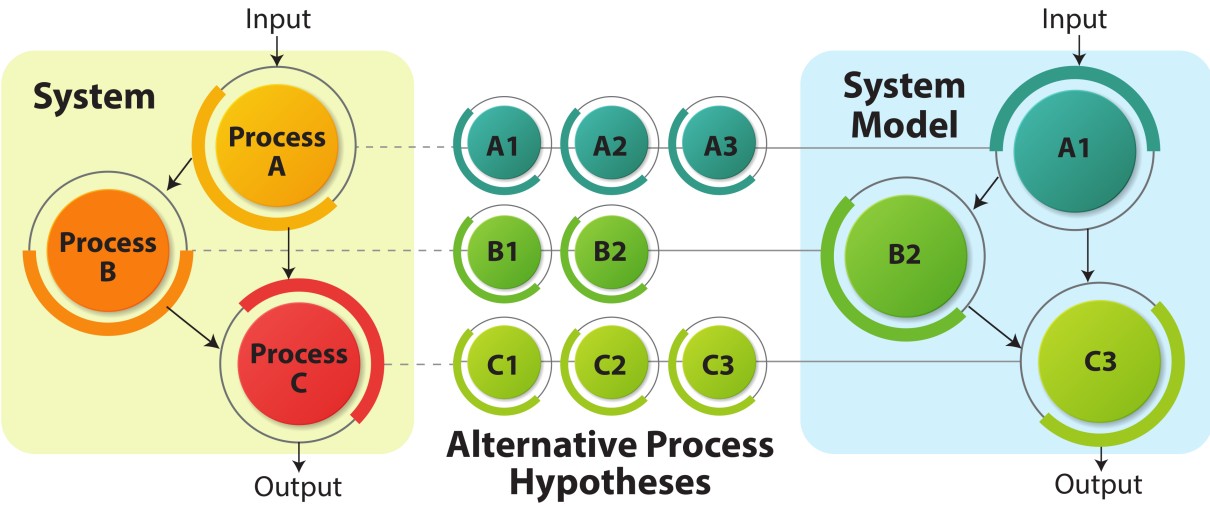

**Figure 1.** Schematic to illustrate a real world system (yellow box) comprised of three-processes (red shapes). Multiple hypotheses or assumptions exist for each process, three for process A, two for process B, and three for process C. When a modeller is building a conventional model of the system (blue box) they are faced with the choice of which hypothesis or assumption to for each process in their model. In this illustration, the model is composed of hypothesis A1 for process A, hypothesis B2 for process B, and hypothesis C3 for process C. MAAT allows a modeller to use all available hypotheses for each process, and compare them using formal and informal methods. In this illustration, a total of 18 possible models exist. The addition of one more process with three alternative representations would increase the number of possible models to 54.

the system model. The system model is highly modular by design, allowing flexible model structure according to information passed from the interface. Algorithms to analyse the sensitivity of model outputs to variation in process representations and parameters are contained within the wrapper.

MAAT is designed to be as system model agnostic as possible. However, our primary domain of research is biogeosciences
5  and ecosystem ecology and MAAT v1.0 comes packaged with a unified multi-assumption leaf-scale photosynthesis model. Photosynthesis is a central process of the biosphere. At the heart of many Terrestrial Ecosystem and Biosphere Models (TBMs) lie the mathematical hypotheses describing the enzyme kinetics of photosynthesis and the hypotheses and assumptions describing associated processes, e.g. stomatal conductance. Enzyme kinetic models lie at the core of TBMs in order to accurately simulate the ecophysiological interaction of terrestrial ecosystems with the interrelated carbon, water, and energy cycles of the
10  Earth System. Many studies have demonstrated the sensitivity of TBM predictions to variation in parameters and assumptions used to represent these core model processes (e.g. Zaehle et al., 2005; Rogers et al., 2017; Anav et al., 2015; Bonan et al., 2011; Walker et al., 2017b).

Variability in numerical model output comes from multiple sources, not solely from uncertain process knowledge. Other sources of model variability are variable or uncertain parameter values, input scenarios (boundary conditions), and initial



conditions (Beven, 2016, 2006; Vrugt et al., 2009). Sensitivity analysis (SA) tests the response of model outputs to pre-defined variation in any of the above-mentioned sources of model variability (Razavi and Gupta, 2015; Song et al., 2015). Parametric uncertainty in models has many established methods for its assessment and quantification. These methods are often based on Monte-Carlo (MC) techniques that run large ensembles of model simulations that sample parameter space, boundary condition

space, and initial condition space (Saltelli et al., 2010; Song et al., 2015; Dai and Ye, 2015). Some formal SA methods exist for assessment of model output sensitivity to variable process representation (e.g. Dai et al., 2017; Green, 1995), and are based on similar MC techniques combined with model averaging. However, methods to assess model sensitivity to variable process representation are few and less extensively used.

To apply parametric SA methods requires a model of the system of interest, a wrapper to sample parameter space and run

the model, and an interface to pass information (often both ways) between the wrapper and the model. As with parametric SA methods, application of process representation SA methods requires a model of the system of interest, a wrapper that samples the configuration of the ensemble member, and an interface to pass information between the wrapper and the model. The practical challenge in developing these methods is to design an interface that enables the model to accept information on which process representations to use, and to configure the model in a way that is computationally efficient. Selecting among

alternative assumptions can be achieved using switches and case (i.e. 'if') statements. However, many large case statements that would be required for extensive process representation variability complicate readability and increase the runtime of the code. The challenge is to represent an assumption simply, as a character string for example, that the system model can interpret to directly access the code that represents the assumption. This also requires a highly modular modelling code.

Most models are not built in this way, though thanks to recent efforts in hydrology we have begun to see models with these

capabilities emerge (Downer and Ogden, 2004; Clark et al., 2015; Coon et al., 2016). In this study we build on these previous efforts by developing a modular modelling code designed explicitly for the generation of large model ensembles that differ in how each process within a system is represented.

**Table 1.** Table of definitions employed in this paper.

| | |
|---|---|
| System | A complex of interconnected and interacting processes. |
| Process | A biological, chemical, or physical mechanism. |
| Hypothesis | A mechanistic description of how a particular process operates. A statement of cause and effect. |
| Model Hypothesis | A mathematical description of a hypothesis (also referred to as representation, process representation, or assumption). |
| Assumption | Anything encoded in a model to represent part of the real world. Used synonymously with process representation. Can include hypotheses, empirical observations of relationships to represent a process that is not fully understood, or a simplification of more detailed mechanism. |





## 2 The Multi-Assumption Architecture and Testbed (MAAT)

MAAT is designed to automate the configuration and implementation of model ensembles with a high degree of flexibility. The ensembles can vary in assumptions and hypotheses (model structure), parameters (functional traits), and boundary conditions (environmental conditions). MAAT is written in R (R Core Team, 2017), which has functions that allow simple and efficient

operation of the code. The prototype style of object-oriented programming (specifically the 'proto' package in R) is used to code the model and the wrapper objects. The 'apply' family of functions are used to run the emsemble and the 'get' function to parse R objects from a character string.

Flexibility and generality is achieved by code modularity. As described in the Introduction, MAAT is composed of a wrapper, an interface, a system model, and alternative process representation functions. The wrapper interprets input data and gener-

ates the model ensemble from those data. Through the interface, the wrapper sequentially passes information for a particular ensemble member to the system model and then runs the model (Fig. 2). The wrapper is a separate object, the system model is a separate object, and the process representations are individual R functions. Each process is a separate function call in the system model code, allowing multiple functions (i.e. hypotheses or assumptions) to represent each process. Different ways to represent the overall system are also separated from the system model object, allowing alternative system conceptualisations

to be incorporated (e.g. light use efficiency versus enzyme kinetic models of photosynthesis). The alternative system functions and process representation functions are called during runtime using character strings, avoiding the use of case statements and model switches. The avoidance of case statements in process specification increases code readability and is especially useful when adding new assumptions for a process, or new processes (by defining new system functions). To add a new assumption, all that must be coded is the function (i.e. no modification of case statements is necessary). This simplicity facilitates rapid

model development and testing of new hypotheses and assumptions.

The modularity of MAAT is such that the wrapper code contains no information that is specific to a particular system model. All information specific to a particular system model is contained with the system model and the input files. Thus the wrapper is completely agnostic to the particularities of the system model. This separation of information allows the development of new system models without the need to alter the wrapper, and with only slight modificiation of the interface.

The two objects fundamental to the operation of MAAT, the wrapper object and the system model object, are now described in greater detail, followed by details on initialisation of MAAT and HPC capabilities.

### 2.1 Wrapper object

The wrapper object manages and runs an ensemble specified by the user. The wrapper object can run an ensemble for a model object that describes any system (provided that the system model is written in the MAAT code formalism). The bottleneck for

application to different systems models is that the model object and associated process functions must be coded in R using the MAAT syntax. This coding is required due to the high degree of modularity of the code, which is not common in existing models. Assuming a model is coded in another language with this hyper modularity, several R functions could be written to call these modules written in other laguages from within MAAT.




The wrapper object contains a data structure, a run function that generates the ensemble and then calls a cascade of run functions that run the ensemble, and output integrating functions. The wrapper is built and called by a run script that also builds and embeds the model object within the wrapper. The run script reads user specified command line arguments and input file(s), interprets this information, and passes it to the wrapper. According to the type of ensemble and analysis specified, the

wrapper integrates input information to generate the ensemble, and then runs the ensemble.

An ensemble is characterised by two things: the variables that vary across an ensemble (called 'dynamic' variables) and the type of ensemble (e.g. factorial, process sensitivity analysis). Variables that do not vary across the entire ensemble are referred to as 'static' variables. Defining the ensemble requires definition of static variables, dynamic variables, their values, and the ensemble type. Static variables and their values are read from a default values file, or specified by the user in the input file. A

user need only provide the static variable values that differ from the defaults and a complete list of all static variable values is not required. Dynamic variables and their multiple values are simply specified by the user in the input file. According to the ensemble type, the wrapper generates the ensemble by combining the dynamic variables into matrices that describe the ensemble with variables in columns and values in rows. These matrices are separate for process representations, parameters, and environment. Finally, and according to ensemble type, the wrapper calls the appropriate run cascade (algorithm) that runs

the specified ensemble type.

The run cascade is a set of functions with a nested call structure that are designed to be called by the 'apply' family of R functions. Each function in the run cascade passes a line of its associated variable matrix to the model configuration function, then calls the next function in the cascade. The final function in the cascade executes the model by calling the model object run function.

## 2.2   Model object

The model object is composed of a data structure, a configuration function (the interface), a run function, process functions (technically these are external to the object but the object will not work without them), output functions, and unit testing functions. The data structure contains multiple lists of variables. Three lists contain the details of the ensemble member, these are: a character string list for each process within the system (labelled 'fnames' in MAAT code), a numerical list of model

parameters (labelled 'pars'), and a numerical list of model boundary conditions (labelled 'env'). These three ensemble member description lists do not vary throughout the run of a single ensemble member. Two additional lists describe the model state at each timestep. These two lists are both numeric and are a list of state variables (labelled 'state') and secondary state variables that can be thought of as dynamic parameters (labelled 'state_pars'). A useful way of thinking about the distinction is that an assumption could be made that these secondary state variables are fixed parameters, while the primary state variables are the

variables intended to be predicted and therefore assuming a fixed value for these would obviate the need for the model.

The configuration function acts as the interface between the wrapper and the model. The 'configure' function is passed values for the three ensemble member description lists by the wrapper object. Each element of the three ensemble member description lists is named. The wrapper passes the configuration function a vector of named values and the configuration functions searches the three lists for named elemnts and assigns values when the elements are found. The object-oriented method and assignment



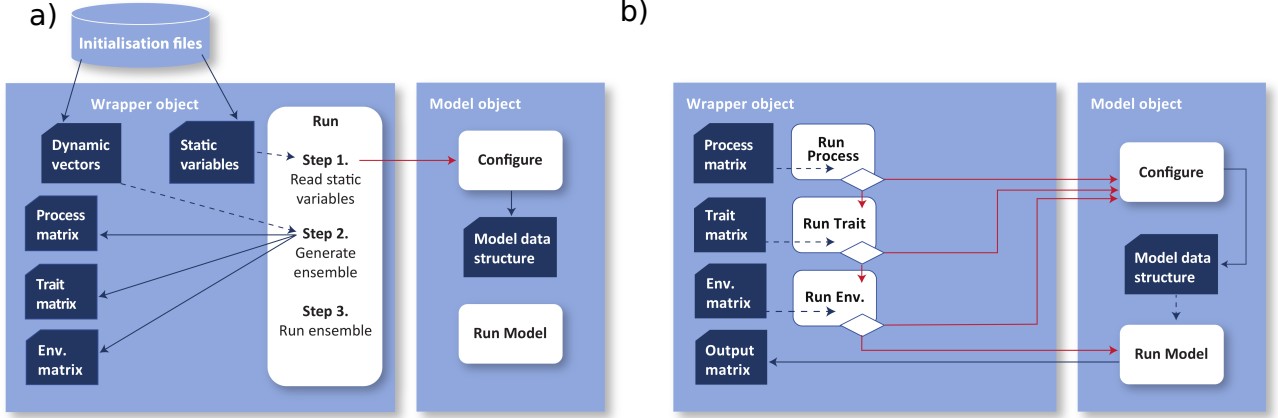

**Figure 2.** Schematic representing the basic structure and execution process of MAAT. Panel a) represents the first two steps of a MAAT execution: 1) reading user input data from initialisation files; and 2) generating ensemble matrices from dynamic variables. Panel b) represents a single iteration of the 'run cascade' which forms the third step of a MAAT execution. 'Proto' objects (light blue boxes) contain data structures (dark blue shapes) and functions (white rectangles). Blue arrows represent the transfer of data via a read (dashed) or write (solid), and red arrows repressent function call. During the execution of the run cascade, each run function associated with a particular variable type (process representation, parameter, environment) reads a line of the variable type matrix and calls the model object configure function with the line from the matrix as an argument. The configure function writes the variable values to the model object data structure, then the run function calls the next run function in the run cascade. The final run function in the cascade is the model run function which executes the model and writes output to the output data structure in the wrapper object.

by variable name provides flexibility in input specification by allowing variable assignment during runtime of only the variables that are varied in the ensemble (called dynamic variables). Variables that do not vary across all ensemble members (called static variables) are assigned by the configuration function at the very beginning of an ensemble simulation. Thus static variable specifications are over-written by dynamic variable specifications. Once the 'configure' function has been called for each of

5   the three ensemble member description lists, the ensemble member has been completely defined and the run function is called by the wrapper.

The 'run' function contains the structure of the system, primarily the order in which the system processes are called and executed. A key component of MAAT's flexibility, and advantage over most other models and modelling frameworks, is that all process hypotheses are written as separate R functions. The process assumption to use for a particular ensemble member

10  is specified using a character string that is the name of the R function that represents that specific hypothesis or assumption. These function name character strings allow the functions to be called using the 'get' function in R, avoiding the need for case statements to select the code to be used to represent a process. All of the process hypothesis functions have an object as their first argument, i.e. the model object that contains parameter and boundary condition values that the function may need to





access. Passing the model object to the function allows for simple argument passing to the functions and relatively clear coding of the system framework.

An output function is written into the model object that allows different combinations of model state and other variables to be output based on an input character string. Unit testing functions are designed to test the operation of the run function under

specific conditions and to compare alternative hypotheses for various process.

### 2.3  Initialisation

An initialisation script is run from the command line and command line arguments can be passed to select various options to define the run. The model to run can be specified as a command line argument; currently only a leaf-scale photosynthesis model and a simple ground water model are available. Any model object coded in the correct format could be provided. The

initialisation script loads the wrapper object and then loads the model object into the wrapper object.

The specifics of the model ensemble are then read by the initialisation script from either standardised R scripts or XML files, specified on the command line. These initialisation files mimic the three lists in the model object data structure, described in the above Section: 'fnames', 'pars', and 'env'. A minimum of two initialisation files are required and read by the initialisation script. The first is the default variable values, an XML file that exactly mimics the three model object lists. This default XML

comes packaged with the source code. The other initialisation file(s) are user defined and contain the static and dynamic variable values for the ensemble. Values to be passed to the wrapper object are specified in these initialisation files and must be named exactly as they appear in the model object data structure.

For the dynamic file, variables can be assigned snippets of R code as a character string, and these will be parsed by the wrapper and the variable assigned the output value of the R code snippet. The use of R code snippets allows variables to be

assigned values that are samples drawn from various distributions of dynamic parameters, with a user defined sample number. The initialisation script also allows some flexibility in the specification of dynamic boundary conditions, such as a time series of meterological data, though the files must be in comma separated ASCII format. The column names of the dynamic boundary condition file are assigned the model object boundary condition names using an XML file similar to the above described files. These dynamic boundary conditions are applied for each ensemble member and are different from the boundary conditions

that are varied as part of the ensemble.

### 2.4  HPC

Due to the large ensembles needed to run global sensitivity analyses, MAAT has been designed to run on High Performance Computing (HPC) systems using the 'mclapply' function from the 'parallel' R package. This package uses the forking method of parallel computing, which relies on shared memory. Therefore MAAT ensembles are currently limited to a single computer

node of multiple cores with shared memory. However, with the current generation of HPC systems that have a large number of cores per node, parallel processing in MAAT can yield substantial increases in speed compared with serial processing. For example, a leaf photosynthesis ensemble with 100 million members runs in around 5 hours on 32 cores with a combined CPU





time of around 172 hours. However, the current requirement for shared memory precludes scalability across nodes of an HPC system and we will return to this in the Discussion.

## 3 Ensemble types

The following Section details the ensemble types that can be generated within MAAT, as well as showing results that verify that the sensitivity analysis algorithm is working as intended.

### 3.1 Factorial combination

The simplest type of ensemble is a complete factorial combination of options. In this case, processes with multiple representations, parameters with multiple values, and environment variables with multiple values are specified in the input file. From these inputs three matrices are configured representing process, parameter, and environment combinations. Each of these matrices is a factorial combination of the values specified for each variable, with variables arranged in columns and their values on the rows. The run cascade in the wrapper object is then called. Each run function in the run cascade passes a row of its associated matrix to the model object configure function and calls the next function in the run cascade (Fig. 2b). The model object configure function places the variable values in the model object data structure. For a factorial simulation the process run function is called first, which then calls the parameter run function, which then calls the environment run function, which then executes the model. On completion of the model execution, the environment run function then passes the next row of the environment matrix to the model and executes the model. This repeats until the last row of the environment matrix is reached, then the parameter run function passes the next row of the parameter matrix to the model object configure function and calls the environment run function again. This is the nested nature of the run cascade and the model is executed for every combination of the lines of the process matrix, parameter matrix, and the environment matrix.

### 3.2 Sensitivity analysis algorithms and verification

Global variance-based sensitivity indices quantify the proportion of variance in model output caused by variability in parameters and processes. Specific ensembles (algorithms) allow the calculation of global parameter sensitivity indices and global process sensitivity indices within MAAT. For global parameter sensitivity analysis the algorithm developed by Saltelli et al. (2010) is employed. As with the parameter sensitivity index, the global process sensitivity index (Dai et al., 2017) accounts for variability in parameters while also accounting for variability caused by different model structures and assumptions, i.e. the different ways in which processes can be represented. The process sensitivity index calculates the proportion of model output variance caused by variation in all of the parameters that feature in a process and by variation in ways in which to represent a process. As an example, in the simplest case one may have two models. Parameter sensitivity can account for the variance in output within each model, but not the variance in model output caused by the two different models themselves (i.e. the difference between the means of the output from the two models). These different components of model output variance can



be thought of as within and between individual model variance. The parameter sensitivity index accounts for within model variance only, while the process sensitivity index accounts for both within and between model variance.

The algorithms for the parameter and the process sensitivity indices are not simply factorial combinations of process representations and parameters (Dai et al., 2017). Therefore the configuration of the 'fnames' and 'pars' matrices and the run cascade is different for each of the algorithms. The algorithms are described in detail in Saltelli et al. (2010) and Dai et al. (2017) so we do not go into detail here. For the Saltelli algorithm, two parameter matrices are constructed, $\mathbf{A}$ and $\mathbf{B}$, with $n$ rows and $n_p$ columns, where $n$ and $n_p$ are the number of samples and the number of parameters in the sensitivity analysis. Each row of these matrices represents a sample from the distributions of each parameter in the analysis. With only a single model, the model is run with each of the parameter sample rows in matrices $\mathbf{A}$ and $\mathbf{B}$. The model is then run again over the $n_p$ $\mathbf{A_B^{(i)}}$ parameter matrices which are the $\mathbf{A}$ matrix with column $i$ replaced with column $i$ from the $\mathbf{B}$ matrix and $1 \leq i \leq n_p$. When multiple models are available, each model is run over matrices $\mathbf{A}$, $\mathbf{B}$, and $\mathbf{A_B^{(i)}}$. As MAAT is designed to switch in alternative assumptions (hypotheses, representations, or structures) for each process in the analysis, the number of all possible models is $\prod_{k=1}^{n_k} \phi_k(j)$ where $n_k$ is the number of processes in the sensitivty analysis and $\phi_k(j)$ is the number of representations of process $k$. With both variable processes and parameters, the total number of iterations in this algorithm is: $(2 + n_p)n \prod_{k=1}^{n_k} \phi_k(j)$.

The process sensitivity algorithm runs once for each of the $n_k$ process in the sensitivity analysis. For each process, $\phi_k$, the algorithm runs each of the $\phi_k(j)$ representations of $\phi_k$ over a parameter matrix $\mathbf{A_{\phi_k}}$. $\mathbf{A_{\phi_k}}$ has $n$ rows and $n_{pk}$ columns where $n$ is the number of samples and $n_{pk}$ is the number of parameters in $\phi_k$. For each of these process representations and parameter samples, the algorithm then runs the $\phi_{\tilde{k}}(j)$ representations of all the other processes in the analysis, $\phi_{\tilde{k}}$ over parameter matrix $\mathbf{A_{\phi_{\tilde{k}}}}$. $\mathbf{A_{\phi_{\tilde{k}}}}$ has $n$ rows and $n_{p\tilde{k}}$ columns where $n$ is the number of samples and $n_{p\tilde{k}}$ is the number of parameters in all processes $\phi_{\tilde{k}}$. The total number of iterations in the process senstivity analysis is: $n_k n^2 \prod_{k=1}^{n_k} \phi_k(j)$.

To verify that the algorithms are working correctly in MAAT we employ the simple groundwater hydrology model presented in Dai et al. (2017). The simple groundwater model calculates hydraulic head across a vertical cross-section of a geographical domain. The model was encoded in MAAT and consists of two processes: recharge and parameterisation of hydraulic conductivity through the underlying geology. Each of these two processes is given two possible representations. For recharge a power law ($R_1$):

$$R_1 = 5.04a(p - 355.6)^{0.5} \tag{1}$$

and a linear model ($R_2$):

$$R_2 = b(p - 399.8) \tag{2}$$

were used, where $a$ and $b$ are scaling parameters and $p$ is precipitation in mm. The second process, parameterisation of hydraulic conductivity through the underlying geology, used a single homogeneous zone representation and a two-zone representation. The parameters varied were a single value of hydrological conductivity ($K$) for the single zone representation and two values





of hydrological conductivity ($K_1$ and $K_2$) for the two-zone model. The study of Dai et al. (2017) ran a parameter and process sensitivity analysis of this simple model assuming that $a$ followed the normal distribution, N(3.35,1) (where 3.35 is the mean and 1 the standard deviation), $b$ the uniform distribution, U(0.1,0.2), $K$ the normal distribution, N(15,1), and $K_1$ and $K_2$ the normal distributions N(20,1) and N(10,1), respectively. Clearly there are other parameters that could have been varied in this

sensitivity analysis, but the analysis was run for illustrative purposes comparing the parameter and process sensitivity indices. The parameter sensitivity indices (Table 2) and process sensitivity indices (Table 3) calculated by Dai et al. (2017) and in this study demonstrate that the MAAT algorithms are operating correctly. Convergence of the calculated process sensitivity index is achieved with an $n$ of around 200 (Fig. 3). Moreover, the large differences in parameter sensitivities depending on model combination clearly demonstrates the need for multi-assumption modelling and tools like MAAT.

**Table 2.** Global parameter sensitivity index ($S_i$) for hydraulic head calculted by the hydrology model described in Dai et al. (2017), calculated using Saltelli's algorithm. Results are presented from Dai and using MAAT in this study, demonstrating the correct implementation of Saltelli's algorithm in MAAT. The slight differences are caused by random sampling.

|       | $R_1G_1$ | | $R_1G_2$ | | $R_2G_1$ | | $R_2G_2$ | |
| $S_i$ | $a$ | $K$ | $a$ | $K_1K_2$ | $b$ | $K$ | $b$ | $K_1K_2$ |
| --- | --- | --- | --- | --- | --- | --- | --- | --- |
| Head (Dai et al. 2017) | 94.8 | 4.8 | 61.5 | 37.8 | 88.7 | 10.6 | 6.5 | 93.2 |
| Head (This study) | 94.8 | 4.9 | 61.5 | 38.3 | 88.7 | 10.8 | 6.6 | 93.4 |

**Table 3.** Global process sensitivity index ($S_k$) for hydraulic head calculated by the hydrology model, calculated using the algorithm described in Dai et al. (2017). Results are presented from Dai and using MAAT in this study, demonstrating the correct implementation of the algorithm in MAAT. As above, the small differences are caused by random sampling.

| $S_k$ | Recharge | Geology |
| --- | --- | --- |
| Head (Dai et al. 2017) | 28.4 | 67.9 |
| Head (This study) | 29.1 | 71.6 |

**4   Unified multi-assumption model of leaf-scale photosynthesis**

In this section we describe the unified, multi-assumption model of leaf-scale photosynthesis, focusing on enzyme kinetic models of photosynthesis (Farquhar et al., 1980; von Caemmerer, 2000). Our intention is to provide a comprehensive review of the various processes and their associated assumptions key to simulating leaf-scale photosynthesis. The inclusion of assumptions is based primarily on the methods used to simulate leaf-scale photosynthesis in TBMs, with some augmentation from common

or more recently defined hypotheses and assumptions.





**Figure 3.** Standard deviation of calculated $S_k$ showing convergence characteristics as a function of sample size. Calculated by resampling and subsampling a single ensemble 10 times for each subsample $n$. Decreasing standard deviation demonstrates convergence on a solution. Dashed lines represent a standard deviation in $S_k$ of 0.01 and 0.001.

In drawing together in a single place and unifying the various hypotheses and assumptions commonly used in physiological models and TBMs, we aim to provide a useful resource for researchers and students alike, in addition to providing a guide to how these processes are simulated in MAAT. In this review and unification we draw upon Farquhar et al. (1980); Collatz et al. (1991); von Caemmerer (2000); Medlyn et al. (2002); Gu et al. (2010), as well as many other references. At times we may

introduce notation that is different from the notation in the original papers. In the few cases where we do change notation, the aim is an attempt to integrate some of the disparate notation in the literature by using the same symbol to refer to common variables. The following sections are arranged by each process within leaf-scale enzyme kinetic models of photosynthesis. Within each section the various competing hypotheses and assumptions are presented in unified definitions and units.

### 4.1 Carbon assimilation

Enzyme kinetic models of leaf photosynthesis (Farquhar et al., 1980; Collatz et al., 1991; von Caemmerer, 2000) simulate net $CO_2$ assimilation ($A$, $\mu mol\, CO_2\, m^{-2}s^{-1}$) as the gross carboxylation rate ($A_g$, $\mu mol\, CO_2\, m^{-2}s^{-1}$) scaled to account for the photorespiratory compensation point ($\Gamma_*$, Pa; the chloroplast $CO_2$ partial pressure at which the carboxylation rate is equal to the rate of $CO_2$ release from oxygenation), minus non-photorespiratory ('day') respiration ($R_d$, $\mu mol\, CO_2\, m^{-2}s^{-1}$):

$$A = A_g(1 - \Gamma_*/C_c) - R_d \tag{3}$$

where $C_c$ is the chloroplast $CO_2$ partial pressure (Pa). $A_g$ is a function of three potentially limiting gross carboxylation rates: the RuBisCO limited rate ($A_{c,g}$); the electron transport limited rate ($A_{j,g}$); and the triose phosphate use limited rate ($A_{p,g}$). We introduce this notation, using $A$ to always refer to carbon assimilation and subscripts as classifiers, in an attempt to integrate some of the disparate notation in the literature. To select the limiting rate, Farquhar et al. (1980) used simply the minimum rate:

$$A_g = min\{A_{c,g}, A_{j,g}, A_{p,g}\} \tag{4a}$$





To be precise, Farquhar et al. (1980) described only the first two limiting rates, but their method can be used to include the third. Collatz et al. (1991) introduced two quadratics to apply non-rectangular hyperbolic smoothing among the potentially limiting rates:

$$0 = \theta_{cjp}A_g^2 - (A_{cj,g} + A_{p,g})A_g + A_{cj,g}A_{p,g} \tag{4b}$$

and

$$0 = \theta_{cj}A_{cj,g}^2 - (A_{c,g} + A_{j,g})A_{cj,g} + A_{c,g}A_{j,g} \tag{4c}$$

where $A_{cj,g}$ is a latent variable, and $\theta_{cjp}$ and $\theta_{cj}$ are smoothing parameters ($\beta$ and $\theta$ in Collatz's original notation). We change the original notation to use $\theta$ for any smoothing parameter with subscripts as classifiers. Simply selecting the minimum rate is a special case of the Collatz et al. (1991) method where $\theta_{cjp}$ and $\theta_{cj}$ are both equal to one.

All potential gross carboxylation rates, $A_{c,g}$, $A_{j,g}$, and $A_{p,g}$, are modelled as Michaelis-Menten functions of $C_c$. For $A_{c,g}$, $V_{cmax}$ ($\mu$mol $CO_2$ m$^{-2}$s$^{-1}$) determines the asymptote:

$$A_{c,g} = \frac{V_{cmax}C_c}{C_c + K_c(1 + O/K_o)} \tag{5}$$

where $O$ is the chloroplast $O_2$ partial pressure (kPa; assumed to be atmospheric $O_2$ partial pressure); $K_c$ and $K_o$ are the Michaelis-Menten constants of RuBisCO for $CO_2$ (Pa) and $O_2$ (kPa). For $A_{j,g}$, the asymptote is the electron transport rate ($J$; $\mu$mol m$^{-2}$ s$^{-1}$) divided by four to represent the four electrons needed to reduce the NADP required for one carboxylation reaction:

$$A_{j,g} = \frac{J}{4}\frac{C_c}{C_c + 2\Gamma_*} \tag{6}$$

For $A_{p,g}$, the asymptote is proportional to the rate of triose phosphate utilisation ($TPU$; $\mu$mol m$^{-2}$ s$^{-1}$):

$$A_{p,g} = \frac{3TPU C_c}{C_c + (1 + 3\alpha_T)\Gamma_*} \tag{7}$$

where $\alpha_T$ represents the fraction of triose phosphate exported from the chloroplast that is not returned. Theoretically, $\alpha_T$ can take values between 0 and 1. In practice, values >1 have been observed (Gu, unpublished), suggesting that $\alpha_T$ may also be

accounting for processes yet to be fully described.

Photorespiration releases a molecule of $CO_2$ for every two oxygenation reactions (catalysis of $O_2$ and ribulose 1,5-bisphosphate by RuBisCO) (Farquhar et al., 1980), therefore oxygenation reduces the net carbon assimilation rate. The $C_c$ partial pressure at which carbon assimilation equals $CO_2$ release from photorespiration is known as the photorespiratory compensation point, $\Gamma_*$, described above. $\Gamma_*$ can be described by the kinetic properties of RuBisCO (Farquhar et al., 1980):

$$\Gamma_* = \frac{K_c O k_o}{2K_o k_c} \tag{8}$$

where $k_c$ and $k_o$ are the respective turnover rates (s$^{-1}$) of RuBisCO for carboxylation and oxygenation. As described by Eq. 8, $\Gamma_*$ is determined by the ratio of these two parameters, $k_o : k_c$, the ratio of RuBisCO's Michaelis-Menten constants, and the





oxygen partial pressure. Collatz et al. (1991) used:

$$\Gamma_* = \frac{O}{2\tau} \tag{9}$$

where $\tau$ is the $CO_2$-$O_2$ specificity ratio of RuBisCO and is equal to $\frac{K_o k_c}{K_c k_o}$. Therefore $k_o : k_c = \frac{K_o}{\tau K_c}$. Bernacchi et al. (2001) introduced an independent $\Gamma_*$, and simply set $\Gamma_*$ as a constant nominal or base rate at a reference temperature.

Many of the biochemical rates described above are determined by enzymes and are therefore sensitive to temperature. Commonly, to model these parameters the rates are determined at a reference temperature and are then scaled using a temperature response function. We return to these in Sections 4.4 and 4.5 below.

### 4.1.1 Electron transport

The electron transport rate ($J$) is a function of incident photosynthetically active radiation ($I$; $\mu$mol m$^{-2}$ s$^{-1}$). A number

of formulations to represent $J$ exist, and the most commonly used are the following three representations. Following Smith (1937), two representations of $J$ saturate at a maximum rate of electron transport rate ($J_{max}$). One, formulated by Harley et al. (1992):

$$J = \frac{a\alpha_i I}{\left[1 + \left(\frac{a\alpha_i I}{J_{max}}\right)^2\right]^{0.5}} \tag{10a}$$

and the other by Farquhar and Wong (1984):

$$0 = \theta_j J^2 + a\alpha_i I J_{max} J + a\alpha_i I J_{max} \tag{10b}$$

where $\theta_j$ is the non-rectangular hyperbola smoothing parameter. Collatz et al. (1991) proposed a linear light response model with no maximum rate:

$$J = a\alpha_i I \tag{10c}$$

where $a$ is the leaf absorptance and $\alpha_i$ is the intrinsic quantum efficiency of electron transport (the product of $a$ and $\alpha_i$ gives the apparent quantum efficiency of electron transport). $\alpha$ has been used with various meaning in the three original papers

describing these three electron transport models. Farquhar et al. (1980) did not use $\alpha$, instead they used $0.5(1 - f)$ where $f$ is the "fraction of light not absorbed by chloroplasts", defining $I$ as the "absorbed photon flux", and 0.5 accounts for the two photons needed to fully transport a single electron to the thylakoid membrane bound NADP reductase. This is the intrinsic quantum efficiency and equivalent to $\alpha_i$ in our notation. Harley et al. (1992) defined $\alpha$ as the "... efficiency of light energy conversion on an incident light basis" which is equivalent to the apparent quantum efficiency, or $a0.5(1 - f)$ using Farquhar

et al. (1980) notation. Collatz et al. (1991) defined $\alpha$ as the "... intrinsic quantum efficiency for $CO_2$ uptake" which is equivalent to $0.5(1 - f)/4$ using Farquhar et al. (1980) notation, and is more correctly referred to as the intrinsic quantum yield.

Our choice of notation lends itself to consistent notation when modelling photosynthesis across leaf and canopy scales because leaf absorptance, $a$, is equivalent to $1 - \sigma$, where $\sigma$ is defined as the leaf scattering coefficient (the sum of light



reflection and transmission) in many canopy radiative transfer schemes (Spitters, 1986; Wang, 2003). However, our notation is at odds with measuring leaf scale photosynthesis as measurements combine $a$ and $\alpha_i$ into a single term, i.e. the apparent quantum efficiency, because leaf light absorptance or reflection and transmission is not quantified. This inconsistency motivates our use of the subscript $i$ on $\alpha_i$. For the unified photosynthesis model in MAAT we avoid confusion over the definition of $\alpha$

and use $f$ as the parameter which determines intrinsic quantum efficiency ($\alpha_i = 0.5(1-f)$). Specifically, $f$ is the fraction of absorbed light not absorbed by the light harvesting complexes, and accounts for light spectral quality and light absorbtion by cell walls etc.

## 4.2  CO$_2$ diffusion and resistance

The partial pressure of $CO_2$ at the site of carboxylation ($C_c$) is simulated as a function of the rate of $CO_2$ assimilation ($A$), the

atmospheric $CO_2$ partial pressure ($C_a$, Pa), and the resistance of the pathway to $CO_2$ diffusion from the atmosphere to the site of carboxylation ($r$; m$^2$s mol$^{-1}$ CO$_2$). This is simulated by Fick's Law, an analogue of Ohm's Law for electrical circuits:

$$C_c = C_a - rAp \tag{11}$$

where $p$ is atmospheric pressure (MPa). Often resistance is presented in terms of its inverse, conductance ($g$). We opt to use resistance as it linearises Eq 11, and the total resistance of a set of resistors in series is simply their sum. $r$ can be broken

down into a number of different components to the resistance pathway—leaf boundary layer resistance ($r_b$; m$^2$s mol$^{-1}$ H$_2$O), stomatal resistance ($r_s$; m$^2$s mol$^{-1}$ H$_2$O), and internal or mesophyll resistance ($r_i$; m$^2$s mol$^{-1}$ CO$_2$):

$$r = 1.4r_b + 1.6r_s + r_i \tag{12}$$

Note that by convention $r_b$ and $r_s$ are in H$_2$O units as they also determine plant water loss and are used in soil-vegetation-atmosphere water transport models which are often built from analogous equations. The scalars, 1.4 and 1.6, represent the

ratios of $CO_2$ to H$_2$O diffusion resistance. Eq. 11 can be broken out for each of the resistance terms:

$$C_b = C_a - 1.4r_bAp \tag{13a}$$
$$C_i = C_b - 1.6r_sAp \tag{13b}$$
$$C_c = C_i - r_iAp \tag{13c}$$

where $C_i$ (Pa) is the $CO_2$ partial pressure in the mesophyll airspaces of the leaf; $C_b$ (Pa) is the leaf boundary layer $CO_2$ partial

pressure.

### 4.2.1  Stomatal conductance

Stomatal resistance is the key process in the diffusion of $CO_2$ from the atmosphere to the site of carboxylation, though in recent years internal resistance has also been the focus of much research. For consistency with the physiological literature (where most stomatal research originates) we present the following stomatal subsection in conductance, noting that the MAAT



**Table 4.** Table of notation.

| Symbol | Unit | Description | |
|---|---|---|---|
| $a_{vn}$ | $\mu mol\,CO_2\,m^{-2}s^{-1}$ | Intercept of $V_{cmax,Tr}$ to leaf N relationship. | Eq. 19a |
| $b_{vn}$ | $\mu mol\,CO_2\,m^{-2}s^{-1}g^{-1}N$ | Slope of $V_{cmax,Tr}$ to leaf N relationship. | Eq. 19a |
| $n_{vn}$ | $\mu mol\,CO_2\,m^{-2}s^{-1}g^{-1}N$ | Normalisation constant of $V_{cmax,Tr}$ to leaf N power-law. | Eq. 19b |
| $e_{vn}$ | - | Exponent of $V_{cmax,Tr}$ to leaf N power-law. | Eq. 19b |
| $a_{jv}$ | $\mu mol\,e\,m^{-2}s^{-1}$ | Intercept of $J_{max,Tr}$ to $V_{cmax,Tr}$ relationship. | Eq. 20a |
| $b_{jv}$ | $e\,CO_2{}^{-1}$ | Slope of $J_{max,Tr}$ to $V_{cmax,Tr}$ relationship. | Eq. 20a |
| $n_{jv}$ | $e\,CO_2{}^{-1}$ | Normalisation constant of $J_{max,Tr}$ to $V_{cmax,Tr}$ power-law. | Eq. 20b |
| $e_{jv}$ | - | Exponent of $J_{max,Tr}$ to $V_{cmax,Tr}$ power-law. | Eq. 20b |
| $a_{tv}$ | $\mu mol\,CO_2\,m^{-2}s^{-1}$ | Intercept of $TPU_{Tr}$ to $V_{cmax,Tr}$ relationship. | Eq. 21 |
| $b_{tv}$ | - | Slope of $TPU_{Tr}$ to $V_{cmax,Tr}$ relationship. | Eq. 21 |
| $a_{rv}$ | $\mu mol\,CO_2\,m^{-2}s^{-1}$ | Intercept of $R_{d,Tr}$ to $V_{cmax,Tr}$ relationship. | Eq. 22a |
| $b_{rv}$ | - | Slope of $R_{d,Tr}$ to $V_{cmax,Tr}$ relationship. | Eq. 22a |
| $a_{rn}$ | $\mu mol\,CO_2\,m^{-2}s^{-1}$ | Intercept of $R_{d,Tr}$ to leaf N relationship. | Eq. 22b |
| $b_{rn}$ | $\mu mol\,CO_2\,m^{-2}s^{-1}g^{-1}N$ | Slope of $R_{d,Tr}$ to leaf N relationship. | Eq. 22b |
| $b_r$ | - | Slope of $R_{d,Tr}$ to $R_{dark,Tr}$ relationship. | Eq. 23 |
| $a_{Q_{10}t}$ | - | Intercept of $Q_{10}$ to leaf temperature relationship. | Eq. 28 |
| $b_{Q_{10}t}$ | $°C^{-1}$ | Slope of $Q_{10}$ to leaf temperature relationship. | Eq. 28 |
| $a_{\Delta St}$ | - | Intercept of $\Delta S$ to previous leaf temperature relationship. | Eq. 29 |
| $b_{\Delta St}$ | $°C^{-1}$ | Slope of $\Delta S$ to previous leaf temperature relationship. | Eq. 29 |
| $a_{jvt}$ | - | Intercept of $b_{jv}$ to previous leaf temperature relationship. | Eq. 30 |
| $b_{jvt}$ | $°C^{-1}$ | Slope of $b_{jv}$ to previous leaf temperature relationship. | Eq. 30 |
| $a$ | - | Leaf absorbtance, proportion of incident light absorbed by leaf. | Eq. 10 |
| $a_T$ | $°C^{-2}$ | Coefficient of quadratic temperature scaling. | Eq. 27 |
| $b_T$ | $°C^{-1}$ | Coefficient of quadratic temperature scaling. | Eq. 27 |
| $c_T$ | - | Coefficient of quadratic temperature scaling. | Eq. 27 |
| $A$ | $\mu mol\,CO_2\,m^{-2}s^{-1}$ | Net carbon assimilation rate. | Eq. 3 |
| $A_g$ | $\mu mol\,CO_2\,m^{-2}s^{-1}$ | Gross (of photo and non-photo respiration) carbon assimilation rate. | Eqs. 3 & 4 |
| $A_{c,g}$ | $\mu mol\,CO_2\,m^{-2}s^{-1}$ | RuBP saturated potential gross carbon assimilation rate. | Eqs. 4 & 5 |
| $A_{j,g}$ | $\mu mol\,CO_2\,m^{-2}s^{-1}$ | RuBP limited potential gross carbon assimilation rate | Eqs. 4 & 6 |
| $A_{p,g}$ | $\mu mol\,CO_2\,m^{-2}s^{-1}$ | TPU limited potential gross carbon assimilation rate. | Eqs. 4 & 7 |
| $A_{cj,g}$ | $\mu mol\,CO_2\,m^{-2}s^{-1}$ | Potential gross carbon assimilation rate once RuBP limitation/saturation has been accounted for. | Eq. 4c |





| Symbol | Unit | Description | |
|---|---|---|---|
| $C_a$ | Pa | Atmospheric $CO_2$ partial pressure. | Eqs. 11, 13, 16, & 18 |
| $C_b$ | Pa | Leaf boundary layer $CO_2$ partial pressure. | Eq. 13 |
| $C_{b,m}$ | µmol $CO_2$ mol | Leaf boundary layer $CO_2$ molar mixing ratio. | Eq. 14, 16, & 18 |
| $C_i$ | Pa | Internal leaf airspace $CO_2$ partial pressure. | Eq. 13 |
| $C_c$ | Pa | Leaf chloroplastic $CO_2$ partial pressure. | Eq. 3, 5, 6, 7, 11, 13, & 16 |
| $D$ | kPa | Leaf boundary layer $H_2O$ vapour pressure deficit. | Eq. 14 |
| $D_0$ | kPa | Vapour pressure deficit scaling parameter. | Eq. 14 |
| $D_*$ | kPa | Vapour pressure deficit scaling parameter related to $D_0$ and $g_{1,l}$. | Eq. 14 |
| $d_l$ | m | is the leaf dimension perpendicular to the wind direction. | Eq. 15 |
| $\boldsymbol{e}$ | - | A vector of variables to which stomatal conductance responds. | Eq. 14, & 18 |
| $f$ | - | Fraction of light absorbed by leaf not absorbed by photosystems. | Eq. 10 |
| $f_0$ | - | Stomatal conductance parameter related to $g_{1,l}$. | Eq. 14 |
| $f_{lnr}$ | - | Fraction of leaf N in RuBisCO. | Eq. 19c |
| $f_{nr}$ | - | Fraction of RuBisCO that is N. | Eq. 19c |
| $g_s$ | mol $H_2O$ m$^{-2}$s$^{-1}$ | Stomatal conductance, inverse of $r_s$. | Eq. 14 |
| $g_0$ | mol $H_2O$ m$^{-2}$s$^{-1}$ | Minimum stomatal (and cuticular) conductance. | Eq. 14 |
| $g_{1,b}$ | %$^{-1}$ | Stomatal conductance slope from Ball et al. (1987). | Eq. 14 |
| $g_{1,l}$ | - | Stomatal conductance slope from Leuning (1990). | Eq. 14 |
| $g_{1,m}$ | kPa$^{-0.5}$ | Stomatal conductance slope from Medlyn et al. (2011). | Eq. 14 |
| $h_r$ | - | Leaf boundary layer relative humidity. | Eq. 14 |
| $H_a$ | J mol$^{-1}$ | Activation energy for biochemical rate. | Eqs. 25a & 26c |
| $H_d$ | J mol$^{-1}$ | Parameter describing decrease of biochemical rate with temperature. | Eq. 26 |
| $I$ | µmol photons m$^{-2}$s$^{-1}$ | Light incident on the leaf. | Eq. 10 |
| $J$ | µmol e m$^{-2}$s$^{-1}$ | Electron transport rate. | Eq. 6 & 10 |
| $J_{max}$ | µmol e m$^{-2}$s$^{-1}$ | Maximum electron transport rate at $T_l$. | Eq. 10 |
| $J_{max,Tr}$ | µmol e m$^{-2}$s$^{-1}$ | Maximum electron transport rate at $T_r$. | Eq. 20 |
| $K$ | Pa | Michaelis-Menten half-saturation parameter(s) from Eqs. 5, 6 & 7. | Eq. 18 |
| $K_c$ | Pa | Michaelis-Menten half-saturation constant for RuBisCO carboxylation. | Eqs. 5 & 8 |
| $K_o$ | kPa | Michaelis-Menten half-saturation constant for RuBisCO oxygenation. | Eqs. 5 & 8 |
| $k_c$ | s$^{-1}$ | Turnover rate for RuBisCO $CO_2$ carboxylation. | Eq. 8 |
| $k_o$ | s$^{-1}$ | Turnover rate for RuBisCO $O_2$ oxygenation. | Eq. 8 |
| $O$ | kPa | Atmospheric $O_2$ partial pressure. | Eqs. 5, 8, & 9 |



| Symbol | Unit | Description | |
|---|---|---|---|
| $N_a$ | $\mathrm{g\,m^{-2}}$ | Leaf N on an area basis. | Eqs. 19 & 22b |
| $p$ | MPa | Atmospheric pressure. | Eqs. 11, 13, & 18 |
| $Q_{10}$ | - | Scalar on biochemical rate for a $10\,^\circ$C increase in temperature. | Eqs. 25b & 28 |
| $R_d$ | $\mathrm{\mu mol\,CO_2\,m^{-2}s^{-1}}$ | Non-photo (day) respiration rate at $T_l$. | Eq. 3 |
| $R_d, Tr$ | $\mathrm{\mu mol\,CO_2\,m^{-2}s^{-1}}$ | Non-photo (day) respiration rate at $T_r$. | Eqs. 22 & 23 |
| $R_dark, Tr$ | $\mathrm{\mu mol\,CO_2\,m^{-2}s^{-1}}$ | Dark adapted (night) respiration rate at $T_r$. | Eq. 23 |
| $R_{sa}$ | $\mathrm{\mu mol\,CO_2\,m^{-2}s^{-1}g^{-1}}$ | RuBisCO specific activity. | Eq. 19c |
| $R$ | $\mathrm{J\,K^{-1}\,mol^{-1}}$ | Universal gas constant. | Eqs. 25 & 26 |
| $r$ | $\mathrm{m^2s\,mol^{-1}\,CO_2}$ | Resistance to $CO_2$ diffusion from the atmosphere to the site of carboxylation. | Eqs. 11 & 12 |
| $r_b$ | $\mathrm{m^2s\,mol^{-1}\,H_2O}$ | Leaf boundary layer resistance to $H_2O$ diffusion from the atmosphere to the leaf boundary layer. | Eqs. 13 & 15 |
| $r_s$ | $\mathrm{m^2s\,mol^{-1}\,H_2O}$ | Stomatal resistance to $H_2O$ diffusion from the leaf boundary layer to the internal leaf air-space. | Eqs. 13 & 14 |
| $r_i$ | $\mathrm{m^2s\,mol^{-1}\,CO_2}$ | Internal/mesophyll resistance to $CO_2$ diffusion from the leaf internal air-space to the site of carboxylation. | Eq. 13 |
| $T_r$ | $^\circ$C | Reference temperature for nominal biochemical rate. | Eqs. 24, 25, & 26 |
| $T_l$ | $^\circ$C | Leaf temperature. | Eqs. 24, 25, & 26 |
| $T_{r,k}$ | K | Reference temperature for nominal biochemical rate. | Eqs. 25a, 26a, & 26c |
| $T_{l,k}$ | K | Leaf temperature. | Eqs. 25a, 26a, & 26c |
| $T_{opt}$ | $^\circ$C | Optimum temperature for biochemical rate. | Eq. 26b |
| $T_{upp}$ | $^\circ$C | Upper temperature parameter for biochemical rate. | Eq. 26d |
| $T_{low}$ | $^\circ$C | Lower temperature parameter for biochemical rate. | Eq. 26d |
| $TPU$ | $\mathrm{\mu mol\,CO_2\,m^{-2}s^{-1}}$ | Triose phosphate utilisation rate at $T_l$. | Eq. 7 |
| $TPU_{Tr}$ | $\mathrm{\mu mol\,CO_2\,m^{-2}s^{-1}}$ | Triose phosphate utilisation rate at $T_r$. | Eq. 21 |
| $t_b$ | $\mathrm{ms^{-0.5}}$ | Turbulent transfer coefficient between the leaf and the air. | Eq. 15 |
| $U$ | $\mathrm{ms^{-1}}$ | Wind speed across the plane of the leaf. | Eq. 15 |
| $V_{cmax}$ | $\mathrm{\mu mol\,CO_2\,m^{-2}s^{-1}}$ | Maximum RuBisCO carboxylation rate at $T_l$. | Eq. 5 |
| $V_{cmax,Tr}$ | $\mathrm{\mu mol\,CO_2\,m^{-2}s^{-1}}$ | Maximum RuBisCO carboxylation rate at $T_r$. | Eqs. 20, 21, & 22a |
| $V$ | $\mathrm{\mu mol\,CO_2\,m^{-2}s^{-1}}$ | Asymptote parameter(s) from Eqs. 5, 6 & 7 | Eq. 18 |

code uses resistance by convention. By adjusting stomatal conductance, $g_s$ ($1/r_s$), a plant can regulate the combined functions of water diffusion out of the leaf and $CO_2$ diffusion into the leaf. Thus, physiological regulation of stomatal conductance is

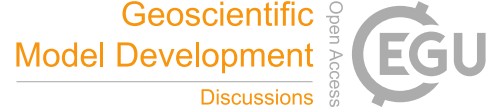



| Symbol | Unit | Description | |
| --- | --- | --- | --- |
| $\alpha_i$ | e photon$^{-1}$ | Intrinsic quantum efficiency, number of electrons transported through the electron transport chain per unit of absorbed light. | Eq. 10 |
| $\alpha_T$ | - | Fraction of exported Triose Phosphate not returned to chloroplast. | Eq. 7 |
| $\Gamma_*$ | Pa | Photorespiratory compensation point, $C_c$ at which $CO_2$ release from photorespiration equals $A_g$. | Eqs. 3, 5–9 |
| $\Gamma$ | Pa | Respiratory compensation point, $C_c$ at which $CO_2$ release from photo and non-photo respiration equals $A_g$. | Eq. 14c, f, & g |
| $\Delta S$ | J mol$^{-1}$K$^{-1}$ | Entropy parameter related to peak of biochemical rate response to temperature. | Eqs. 26a–c |
| $\theta_{cj}$ | - | Non-rectangular hyperbolic smooting parameter for $A_{c,g}$ and $A_{j,g}$. | Eq. 4b |
| $\theta_{cjp}$ | - | Non-rectangular hyperbolic smooting parameter for $A_{cj,g}$ and $A_{p,g}$. | Eq. 4c |
| $\theta_j$ | - | Non-rectangular hyperbolic smooting parameter for electron transport. | Eq. 10b |
| $\kappa_r$ | m$^3$mol$^{-1}$ | is a conversion factor for resistance expressed in s m$^{-1}$ to m$^2$s mol$^{-1}$. | Eq. 15 |
| $\rho_r$ | variable | Nominal biochemical rate at reference temperature. | Eq. 24 |
| $\rho_l$ | variable | Biochemical rate at leaf temperature. | Eq. 24 |
| $\sigma$ | - | Scaling parameter for biochemical rate temperature response. | Eq. 26d |
| $\tau$ | - | $CO_2$-$O_2$ specificity ratio of RuBisCO. | Eq. 9 |
| $\chi$ | - | $C_i$:$C_b$ ratio. | Eq. 14e |

a key process that couples carbon and water cycles from local to global scales (e.g., Medlyn et al., 2011; De Kauwe et al., 2013; Swann et al., 2016). Carbon gain is of benefit to a plant while water loss is a cost in water limited environments, which has led to a large body of research and multiple equations that describe how plants might adjust $g_s$ to balance this conflict. In this section we focus primarily on equations derived from optimisation theory and empirical data that are used in TBMs, recognising that this is not a complete list of all hypotheses on stomatal conductance in the literature (e.g. Buckley et al., 2016; Wolf et al., 2016).

A general form for many stomatal conductance equations, especially those commonly used in TBMs, is:

$$g_s = g_0 + f(\boldsymbol{e})\frac{A}{C_{b,m}} \tag{14a}$$





where $A$ is net carbon assimilation; $f(e)$ is a function of various environmental variables, often a metric of atmospheric dryness and a slope parameter ($g_1$) describing the change in stomatal conductance in response to a change in $e$; and $g_0$ is the minimum $g_s$ primarily due to cuticular conductance. $C_{b,m}$ is $C_b$ in molar units ($\mu$mol mol$^{-1}$; $C_{b,m} = C_b/p$).

Perhaps the most common form of stomatal conductance used by TBMs is that of Ball et al. (1987):

$$g_s = g_0 + g_{1,b} h_r \frac{A}{C_{b,m}} \tag{14b}$$

where $h_r$ is relative humidity (%) and $g_{1,b}$ is the $g_1$ specific to this formulation. Due to different $f(e)$ functions and environmental variable used $g_1$ does not take the same value for all $g_s$ formulations.

Also used by some TBMs is the formulation by Leuning (1990):

$$g_s = g_0 + \frac{g_{1,l}}{(1 - \Gamma/C_b)(1 + D/D_0)} \frac{A}{C_{b,m}} \tag{14c}$$

where $\Gamma$ is the $CO_2$ compensation point in the presence of both photo and non-photo respiration (Pa); $D$ is vapour pressure deficit (kPa); $D_0$ is $D$ at which $g_s$ is reduced by half, and $g_{1,l}$ is the $g_1$ specific to this formulation.

Based on the two above, semi-empirical models have been followed more recently with a function derived from optimisation theory (Medlyn et al., 2011):

$$g_s = g_0 + \left(1 + \frac{g_{1,m}}{\sqrt{D}}\right) \frac{A}{C_{b,m}} \tag{14d}$$

We will present two more empirical assumptions related to stomatal conductance that are commonly employed in TBMs. These assumptions are based on observations that the $C_i$:$C_a$ ratio is often well conserved. These assumptions do not include a $g_0$ term and assume zero leaf boundary layer resistance, which allows an analytical solution to solving these equations (described in Section 4.3). The first of these assumptions, presented in Prentice et al. (1993) and used in the Lund-Potsdam-Jena (LPJ) family of TBMs, is that $C_i$:$C_a$ is constant, often refered to as $\chi$. Assuming that leaf boundary layer resistance of zero means $C_b$ is equal to $C_a$ and substituting $\chi$ into Eq. 13b gives:

$$g_s = \frac{1.6}{1 - \chi} \frac{A}{C_{b,m}} \tag{14e}$$

Cox et al. (1998) derived an alternative formulation from the Leuning model based on work of Jacobs (1994), and employed in the Joint UK Land Environment Simulator (JULES):

$$\frac{C_i - \Gamma}{C_b - \Gamma} = f_0(1 - D/D_*) \tag{14f}$$

where $f_0 = 1 - 1.6/g_{1,l}$ and $D_* = D_0(g_{1,l}/1.6 - 1)$. Rearranging and substituting Eq. 14f into Eq. 13b gives:

$$g_s = \frac{1.6}{1 - \Gamma/C_b - f_0(1 - \Gamma/C_b)(1 - D/D_*)} \frac{A}{C_{b,m}} \tag{14g}$$





### 4.2.2 Boundary layer and internal resistance

While stomatal resistance is the process that receives the majority of attention from ecophysiologists, boundary layer resistance and internal resistance are also important terms in the resistance pathway of $CO_2$ into the leaf and $H_2O$ out of the leaf. $r_b$ determines the coupling of the leaf with the atmosphere in the canopy boundary layer, and influences the leaf energy balance.

The strength of this coupling determines how different leaf temperatures can be from air temperature, with highly coupled leaves showing the smallest differences between leaf and air temperatures. The magnitude of this coupling and its relationship to leaf heat or cold stress have been shown to be a driver of leaf size globally (Wright et al., 2017). $r_b$ is commonly simulated as a function of leaf size and wind speed (Oleson et al., 2013):

$$r_b = t_b(U/d_l)^{0.5}\kappa_r \tag{15}$$

where $t_b$ is the turbulent transfer coefficient between the leaf and the air ($\mathrm{ms^{-0.5}}$), $U$ is wind speed across the plane of the leaf ($\mathrm{ms^{-1}}$), $d_l$ is the leaf dimension in the wind direction (m), and $\kappa_r$ converts resistance expressed in $\mathrm{sm^{-1}}$ to $\mathrm{m^2smol^{-1}}$.

   Internal resistance, often also referred to as mesophyll resistance, is a composite of multiple resistances (see Evans et al., 2009, for a detailed description of these various components). The response of $r_i$ is under investigation and has been shown to respond to temperature (von Caemmerer and Evans, 2014), light (Campany et al., 2016), and $CO_2$ (Kolbe and Cousins,

2018). While $r_i$ and its environmental responses are active areas of research, most TBMs do not explicitly include mesophyll resistance as a process. The absence of explicit inclusion is because $r_i$ is implicit in most measurements of biochemical rate parameters, especially $V_{cmax}$ and $J_{max}$. Explicit inclusion of $r_i$ would also require these 'apparent' biochemical rates to be modified to their absolute rates. Given the large body of research on 'apparent' biochemical rates and the diversity of $r_i$ responses that are not yet fully understood, TBMs are likely to maintain the stauts quo and implicitely account for $r_i$ in the

near future. For this reason, we only include $r_i$ as a parameter which, by default, is set to zero. However, investigation of the impact of $r_i$ is possible within MAAT and should researchers be interested in evaluating the impact of various relationships of $r_i$ to environment, they would be relatively trivial to incorporate.

### 4.3 Numerical and analytical solution

Eqs 3, 11, and 14a are a system of simulataneous equations with three inter-dependent unknowns, $A$, $r_s$, and $C_c$, that need

solving for $A$. In MAAT, these equations are combined into a single function (called the solver function in MAAT, more formally this is a residual function for which a numerical solver finds the root) and are solved using the 'uniroot' function in R's base package, which is based on the Brent solver. The Brent solver has been shown to be robust in solving these simultaneous equations (Tang, unpublished). MAAT also contains a solver function that assumes $r_s$ is zero, thus allowing a calculation of the magnitude of stomatal limitation on carbon assimilation.

A number of TBMs make three simplifying assumptions to the above described set of simultaneous equations such that $A$ can be solved using a simple analytical solution. The first and second simplifying assumptions are that $r_b$ and $r_i$ are zero (to be accurate, most TBMs assume that $r_i$ is zero). These assumptions mean that $C_b = C_a$, $C_c = C_i$, and that Eq 12 collapses so that





$r = 1.6r_s$. With these assumptions, Eq 11 is identical to Eq 13b. The third simplifying assumption is that $g_0$ is zero. Making these assumptions allows $A$ to cancel when Eq 14a is substituted into Eq 13b, yielding an equation for $C_c$ that is independent of $A$:

$$C_c = C_a \left( 1 - \frac{1.6}{f(\boldsymbol{e})} \right) \tag{16}$$

Eq. 16 and the unified expression of $g_s$ models in Section 4.2.1 allows the analysis of the impact of these simplifying assumptions across all the stomatal conductance models presented in Section 4.2.1.

An analytical solution that makes only the first and second assumptions can also be derived to form a quadratic equation:

$$0 = aA^2 + bA + c \tag{17}$$

where:

$$a = p \left[ 1.6 - \frac{f(\boldsymbol{e})}{C_{b,m}} (C_a + K) \right] \tag{18a}$$

$$b = \left[ -g_0(C_a + K) + p\frac{f(\boldsymbol{e})}{C_{b,m}} \big( V(C_a - \Gamma_*) - R_d(C_a + K) \big) + 1.6p(R_d - V) \right] \tag{18b}$$

$$c = g_0 \big[ V(C_a - \Gamma_*) - R_d(C_a + K) \big] \tag{18c}$$

where $V$ and $K$ are the asymptote and half-saturation parameters of Eqs. 5, 6, and 7 depending on which limiting rate is being calculated. We found that the larger root to the quadratic was the solution for A.

A cubic solution that requires no simplifying assumptions is also possible (Baldocchi, 1994; Yin and Struik, 2009). However, the cubic solution is rarely employed by TBMs as it is not always clear which root provides the correct solution. For the sake of brevity we do not include the cubic solution here.

### 4.4 Nominal biochemical rates

Many of the biochemical rates presented in Section 4.1 are enzymatically controlled and are therefore temperature sensitive.
Commonly these rates are presented normalised to a nominal rate at a common reference temperature often, but not always, 25 °C. In this section we describe the methods used to set various nominal biochemical rates at a reference temperature. In Section 4.5 we present methods used to scale these rates from reference temperatures to leaf temperature. The simplest method to set these nominal rates is to define them as input parameters that do not vary during the course of the simulation, and this is possible in MAAT. Also included are a number of functions which describe relationships among the various biochemical traits, 25  primarily with leaf nitrogen on an area basis ($N_a$; g m$^{-2}$) or in relation to ($V_{cmax}$). In the following functions we use $a$ and $b$ to refer to the intercept and slope of a linear relationship and $n$ and $e$ to refer to the normalisation constant and exponent in a power law relationship (i.e. the intercept and slope respectively of a linear relationship of log transformed variables). We use subscripts to identify the relationships to which these parameters belong (see Table 4 for reference).





### 4.4.1 Vcmax

$V_{cmax}$ is the maximum rate of carboxylation by the enzyme RuBisCO. The N content of RuBisCO in a leaf contributes a substantial proportion of total leaf N (Evans, 1989). Therefore, $V_{cmax}$ is often simulated as an empirical function of leaf N, either as a linear relationship (e.g. Harley et al., 1992):

$$V_{cmax,T_r} = a_{vn} + b_{vn}N_a \tag{19a}$$

or a power law relationship that results from a linear regression of log transformed variables (e.g. Walker et al., 2014):

$$V_{cmax,T_r} = n_{vn}N_a{}^{e_{vn}} \tag{19b}$$

or as a linear relationship with parameters that have more physiological meaning (e.g. Oleson et al., 2010):

$$V_{cmax,T_r} = f_{lnr}f_{nr}R_{sa}N_a \tag{19c}$$

where $f_{lnr}$ is the fraction of leaf N invested in RuBisCO; $f_{nr}$ is the fraction of RuBisCO that is N; and $R_{sa}$ is the specific activity of RuBisCO (i.e. the carboxylation rate per gram RuBisCO; $\mu$mol $CO_2$ g$^{-1}$ RuBisCO).

Alternative methods and hypotheses for predicting $V_{cmax}$ exist, such as the co-ordination hypothesis (Chen et al., 1993; Maire et al., 2012); optimisations constrained by co-ordination, leaf N partitioning, and empirical relationships (i.e. LUNA Ali et al., 2016); and empirical relationships to environment (Verheijen et al., 2013). For a more in depth discussion and evaluation of these various methods see Walker et al. (2017b). Currently MAAT only employs the $V_{cmax}$ assumptions that are represented with explicit functions above.

### 4.4.2 Jmax

Commonly $J_{max}$ is simulated as an empirical function of $V_{cmax}$. This is because the relationship between these two photochemical rates is tight (Wullschleger, 1993; Wohlfahrt et al., 1999; Walker et al., 2014), especially considering the common level of variation in other trait-trait relationships. Commonly employed is the classic linear relationship of Wullschleger (1993):

$$J_{max,T_r} = a_{jv} + b_{jv}V_{cmax,T_r} \tag{20a}$$

often, though with a zero intercept (e.g. Medlyn et al., 2002). More recently, Walker et al. (2014) presented evidence that showed the relationship may be better described by a power law:

$$J_{max,T_r} = n_{jv}V_{cmax,T_r}{}^{e_{jv}} \tag{20b}$$

### 4.4.3 TPU

Triose phosphate utilisation is commonly set as a linear function of $V_{cmax}$:

$$TPU_{T_r} = a_{tv} + b_{tv}V_{cmax,T_r} \tag{21}$$





with the intercept commonly set to zero and the slope to 1/6. Given Eq. 3, Eq. 7, and $\alpha_T$, the slope value of 1/6 is equivalent to the value of TPU given in Collatz et al. (1991).

### 4.4.4 Rd

Commonly leaf daytime respiration is simulated as a linear function of either $V_{cmax}$:

$$R_{d,T_r} = a_{rv} + b_{rv}V_{cmax,T_r} \tag{22a}$$

or leaf N:

$$R_{d,T_r} = a_{rn} + b_{rn}N_a \tag{22b}$$

As a function of $V_{cmax}$, respiration is commonly simulated with zero intercept. Also of interest is that $R_d$ is often observed to be smaller during the day, or in the light, when compared with $R_d$ in dark conditions. The processes that result in the reduction of $R_d$ in the light are not clear and there is some discussion surrounding potential bias in the measurement of how $R_d$ changes when conditions go from light to dark. For a comprehensive review of these discussions see Farquhar and Busch (2017) and Tcherkez et al. (2017). A fixed ratio of $R_d$ to respiration in the dark $R_{dark}$ can be selected:

$$R_{d,T_r} = b_r R_{dark,T_r} \tag{23a}$$

$b_r$ can be simulated as a function of incident light intensity following Brooks and Farquhar (1985) and popularised by Lloyd et al. (1995):

$$
\begin{aligned}
b_r &= 1, & 0 \leq I \leq 10 \\
b_r &= (0.5 - 0.05ln\{I\}), & 10 < I
\end{aligned}
\tag{23b}
$$

### 4.5 Temperature scaling

A number of hypotheses and assumptions exist to describe the instantaneous temperature scaling of the above-described biochemical rates. Rate increases with temperature are usually described with an exponential function. And commonly for respiration, a monotonic increase with temperature is all that is considered. For the other three rates, a decrease with higher temperatures is often also observed. Often in the literature the increase and decrease with temperature are presented as a single function. However, the terms that describe an increase with temperature and a decrease with temperature can often be separated and some of the diversity of temperature scaling comes from mixing separate assumptions on the increase and decrease with temperature.

Instantaneous temperature scaling is an immediate metabolic response. Plants also respond to temperature variation over timescales of days to weeks, commonly referred to as acclimation. These acclimatory temperature responses are commonly represented by describing some of the parameters in the instantaneous response as a function of mean temperatures experienced by the leaf over a pre-defined period. In the following sub-sections we first present hypotheses and assumptions for instantaneous temperatures scaling, then for longer-term acclimation of the temperature response.



### 4.5.1 Instantaneous temperature scaling

All hypotheses and assumptions in this Section are presented as functions of leaf temperature ($T_l$, °C) and reference temperature ($T_r$, °C; i.e. the temperature at which the nominal base rate is measured or calculated, decribed in Section 4.4 apply). The result of all the functions is a scalar such that the product of the scalar and the rate at the nominal temperature ($\rho_r$) gives the rate at leaf temperature ($\rho_l$):

$$\rho_l = \rho_r f(T_l, T_r) \tag{24a}$$

In many cases the function to calculate the scalar can be decomposed into a component that increases with temperature and a component that decreases as temperature increases:

$$f(T_l, T_r) = f_i(T_l, T_r) f_d(T_l, T_r) \tag{24b}$$

The two commonly used scalar functions that increase with temperature are the Arrhenius equation:

$$f_i(T_l, T_r) = exp\left\{ \frac{H_a(T_{l,k} - T_{r,k})}{R T_{l,k} T_{r,k}} \right\} \tag{25a}$$

and the Q10 function:

$$f_i(T_l, T_r) = Q_{10}^{\frac{T_l - T_r}{10}} \tag{25b}$$

where $H_a$ is the activation energy (J mol$^{-1}$); $exp$ is the exponential function; the subscript $k$ refers to temperature in Kelvin (K); R is the universal gas constant (8.31446, J mol$^{-1}$K$^{-1}$ ); and $Q_{10}$ is the factor by which $\rho_l$ increases for each 10 °C increase in $T_l$.

In some cases, and for some variables (e.g. $R_d$), simply increasing with temperature is often all that is assumed and $f(T_l, T_r)$ is equal to $f_i(T_l, T_r)$. However, for some rates there is a decrease associated with higher temperatures. A commonly used function for the decrease is a modification on the Arrhenius equation (Medlyn et al., 2002; Kattge and Knorr, 2007):

$$f_d(T_l, T_r) = \frac{1 + exp\left\{ \frac{T_{r,k}\Delta S - H_d)}{R T_{r,k}} \right\}}{1 + exp\left\{ \frac{T_{l,k}\Delta S - H_d)}{R T_{l,k}} \right\}} \tag{26a}$$

where $H_d$ discribes the decrease with temperature (J mol$^{-1}$), as does $\Delta S$ (J mol$^{-1}$K$^{-1}$) which is referred to as an entropy term (Medlyn et al., 2002). $\Delta S$ and $H_d$ are related to the optimum temperature ($T_{opt}$) where $\rho_l$ is at its maximum:

$$T_{opt} = \frac{H_d}{\Delta S - R ln\left\{ \frac{H_a}{H_d - H_a} \right\}} \tag{26b}$$

A simplified form of Eq 26a was introduced in Collatz et al. (1991):

$$f_d(T_l, T_r) = \frac{1}{1 + exp\left\{ \frac{T_{l,k}\Delta S - H_d)}{R T_{l,k}} \right\}} \tag{26c}$$





And another alternative was introduced in Cox et al. (1998):

$$f_d(T_l, T_r) = \frac{1}{[1 + exp\{\sigma(T_l - T_{upp})\}][1 + exp\{\sigma(T_l - T_{low})\}]} \tag{26d}$$

where $\sigma$ is a scaling exponent; and $T_{upp}$ and $T_{low}$ represent high and low leaf temperatures that bound the temperature response.

Brooks and Farquhar (1985) introduced a quadratic function for scaling $\Gamma_*$ with temperature, which we here modify to result in a scalar:

$$f(T_l, T_r) = 1 + b_T(T_l - T_r) + a_T(T_l - T_r)^2/c_T \tag{27}$$

The quadratic function combines both the ascending and descending component of the temperature response.

### 4.5.2 Acclimation of instantaneous temperature scaling

To allow for acclimation to past temperatures, parameters in the above equations can be assumed as functions of past temperature. Tjoelker et al. (2001) demonstrated a variable respiration $Q_{10}$ that was a linear function of leaf maintenance temperature $\overline{T_l}$:

$$Q_{10} = a_{Q_{10}t} + b_{Q_{10}t}\overline{T_l} \tag{28}$$

and Kattge and Knorr (2007) showed that $\Delta S$ is also a linear function of past leaf temperature:

$$\Delta S = a_{\Delta St} + b_{\Delta St}\overline{T_l} \tag{29}$$

In both of these cases, the slope was negative and both $Q_{10}$ and $\Delta S$ decrease with temperature indicating that the sensitivity to instantaneous temperature increase is lower as plants experience higher temperatures. The decrease in $\Delta S$ with past temperature also indicates that $T_{opt}$ increases with temperature. In addition to modifying temperature scaling parameters, Kattge and Knorr (2007) noticed that temperature acclimation also changed the slope of a linear $J_{max}$ to $V_{cmax}$ relationship:

$$b_{jv} = a_{jvt} + b_{jvt}\overline{T_l} \tag{30}$$

The slope of this function is also negative, indicating a decrease in $J_{max}$ relative to $V_{cmax}$ at higher temperature. Currently in MAAT, $\overline{T_l}$ is simply the leaf temperature, representing steady-state acclimation.

## 5 Multi-assumption photosynthesis code verification

In this section we present the results from some simulations with MAAT. The purpose of these simulations is to verify that the photosynthesis code is working as intended, not to test various implementations against data which we will save for extensive evaluations in future research. The use of both numerical and analytical solutions to the system of simulataneous equations for photosynthesis, as well as multiple instances of stomatal conductance equations (with some designed for analytical solution), provides a testbed for code verification. We also demonstrate a simple comparison among the temperature response functions.



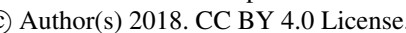



**Figure 4.** Comparison of carbon assimilation against (a) atmospheric $CO_2$ ($A$-$C_a$) curves and (b) internal $CO_2$ ($A$-$C_i$) curves, produced by the simple analytical solution (blue points and lines), the quadratic analystical solution (red points and lines), and the numerical solution (black crosses); for five different representations of stomatal conductance Eqs. 14b–g and two values of $g_0$ (0.00 and 0.01 $molH_2Om^{-2}s^{-1}$).





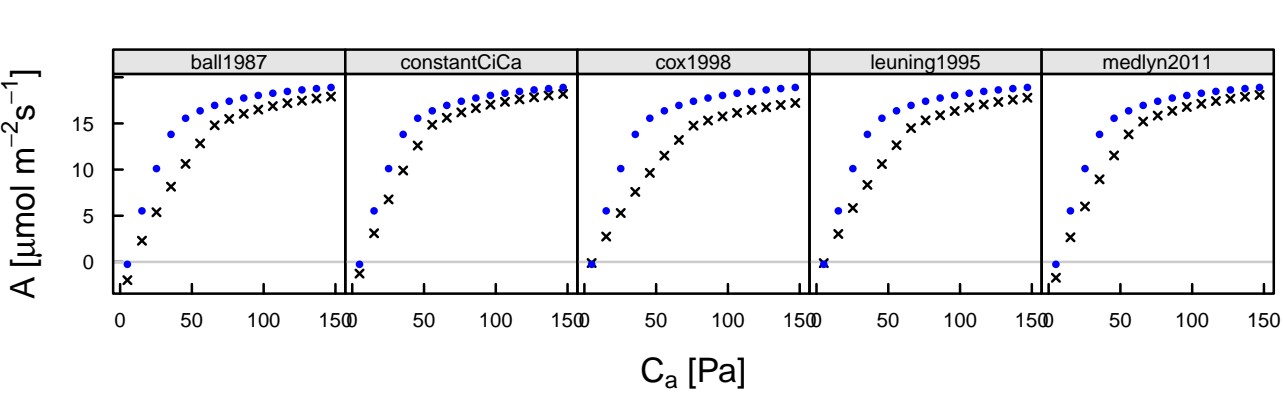

**Figure 5.** Comparison of $A$-$C_a$ curves with and without stomatal resistance (limitation) to carbon assimilation for the five representations of stomatal conductance. $g_0$ equal to 0.01.

## 5.1 Verification of photosynthesis solver

Using both the numerical solution and the simple analytical solution should provide the exact same solutions for carbon assimilation when $g_0$, $r_b$, and $r_i$ are assumed zero. For stomatal conductance hypotheses that include a $g_0$ term, the numerical solution should provide carbon assimilation rates slightly higher than the simple analytical solution because a non-zero $g_0$

slightly decreases resistance to $CO_2$ transport and increases the $C_i$:$C_a$ ratio. Using both the numerical solution and the quadratic analytical solution should provide the exact same solutions when only $r_b$, and $r_i$ are assumed zero.

Fig. 4 shows net carbon assimilation against atmospheric $CO_2$ partial pressure ($A$-$C_a$ curves) calculated using the analytical approximation and full numerical solution with five different representations of stomatal conductance and two values of $g_0$. As described above, when $g_0$ is zero the analytical approximations and the numerical solution should yield the same results. The

top row of panels in Fig. 4a demonstrates this to be the case. When $g_0$ equals $0.01 \ \mathrm{molH_2Om^{-2}s^{-1}}$ the stomatal conductance representations developed to provide a simple analytical solution (Prentice et al., 1993; Cox et al., 1998) again demonstrate equivalence between the analytical approximation and the numerical solution (Fig. 4a). The quadratic solution and numerical solution for the semi-empirical or derived from optimality stomatal conductance representations (Ball et al., 1987; Leuning, 1990; Medlyn et al., 2011) both show a slight increase in $A$ compared with the simple analytical solution because stomatal

conductance is higher when $g_0$ is greater than zero.

MAAT also includes some additional diagnostic tools that can be used to verify the results of the photosynthesis code, and to analyse photosynthesis more broadly. These tools include calculation of the transition point, the value of $C_c$ at which $A_{c,g}$ and $A_{j,g}$ are equal. Plotting the transition point ($C_{c,tran}$), which can be calculated analytically by:

$$C_{c,tran} = \frac{8\Gamma_* V_{cmax}/J_{max} - K_m}{1 - 4V_{cmax}/J_{max}} \tag{31}$$

on the curves (Fig. 4b) also demonstrates that the analytical and numerical solutions are finding the correct transition point.





Another tool can be used to calculate photosynthesis assuming zero total resistance to $CO_2$ transport, $r$, or assuming zero stomatal resistance to $CO_2$ transport, $r_s$. Fig. 5 shows $A$-$C_a$ curves calculated with the numerical solution and $g_0$ equal to 0.01. It is clear from these plots that resistance to $CO_2$ diffusion to the site of carboxylation has a much larger influence on $A$ when the carboxylation rate is limiting compared with when the electron transport rate is limiting.

## 5.2 Temperature response functions

Here we show the various temperature scaling assumptions as an illustration of the decomposition into ascending and descending components and as a simple illustration of MAAT's capability. It is not our intention here to rigorously investigate the effect of parameters and modelling assumptions on the scalar. The ascending and descending components of temperature response functions tend not to be presented separately. However, for a clear demonstration of the difference among the various assumptions, we present the ascending (Fig. 6a), descending (Fig. 6b), and combined (Fig. 6c) temperature response functions over the range 0-45 °C. Some of the assumptions share parameters while others do not. $H_a$ and $T_{opt}$ parameter values were manually adjusted to make the curves as similar as possible and highlight primarily structural differences among the assumptions. This calibration aligned the ascending curves and the peak (maximum) of the temperature response.

Fig. 6a shows that the $Q_{10}$ and Arrhenius relationships can be made to match pretty well, though the Arrhenius relationship gives slightly higher values at the extremes of the temperature range due to the slighly higher base. The descending component of the temperature response shows some slight differences. Collatz et al. (1991) and Cox et al. (1998) preserve the scalar at a value of one for the majority of temperatures below the nominal or reference temperature. However, they also do not preserve $f_d$ at one at the nominal temperature, they both give lower values; 0.95 and 0.96 respectively. The modified Arrhenius equation is the only function that preserves $f_d$ at one at the nominal temperature. However, it does this by having values of $f_d$ above one for temperatures below the nominal temperature; 1.06 at 0 °C. This effect is known and is why activation energy is often given the notation $E_a$ in the Arrhenius equation but is given the notation $H_a$ in the modified Arrhenius equation. $H_a$ is related to the activation energy but is not strictly the activation energy. Not shown in Fig. 6b is that at low temperatures the Cox et al. (1998) assumption can allow a substantial decrease in the scalar (e.g. when $T_{low}$ is 0°C; for this simulation $T_{low}$ was set to -20 °C). Above the reference temperature, the three assumptions show similar declines with the Cox et al. (1998) formulation declining at slightly greater rate.

The differences in the ascending and descending components are reflected in the composite temperature responses (Fig. 6c). The modified Arrhenius assumption has higher values at intermediate temperatures while the the Cox et al. (1998) values are lower at high temperature. The scalar from the Collatz et al. (1991) assumption shows the lowest peak value. While some differences in the scalar are apparently caused by different assumptions, the similarity between the curves suggests that parameter values are likely to be more influential than the specific formulation chosen. However, it is also apparant that parameter values are not entirely interchangeable across assumptions, and that by chosing different assumptions without proper calibration of parameters is likely to lead to substantial differences in the value of the scalar.





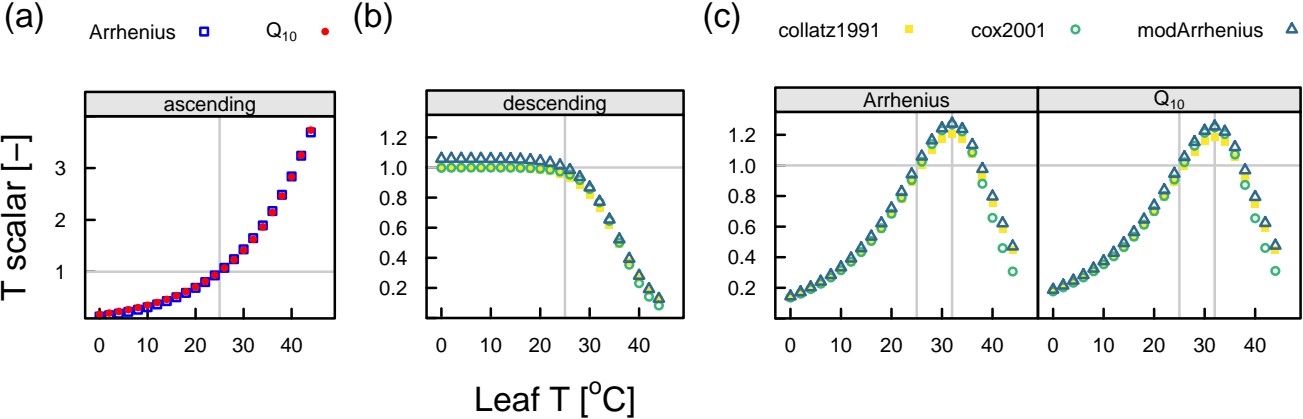

**Figure 6.** Biochemical rate scalars for instantaneous temperature responses, for (a) the ascending component of the response, (b) the descending component of the response, and (c) the combined response. Arrhenius shown as blue squares (a) and on the left panel (c). $Q_{10}$ shown as red circles (a) and on the right panel (c). Descending components (b & c) Collatz et al. (1991) shown as yellow squares, Cox et al. (1998) as green circles, and the modified Arrhenius relationship as blue triangles.

## 6   Discussion

Mathematical computer models are used widely across many scientific domains and industries, primarily for two general purposes: 1) interpreting observations and, 2) making predictions about the piece of the real-world that the model is intended to represent. These two modelling purposes are succinctly summarized by Rastetter (2017) as modelling for understanding and modelling for numbers (i.e. prediction). With the aim of deepening our understanding of competing assumptions and targeting uncertainty reduction in model predictions, we have developed and built a set of software codes, the Multi-Assumption Architecture and Testbed (MAAT v1.0). MAAT facilitates the building and detailed analysis of systems models when there are multiple assumptions (mechanistic hypotheses and empirical or simplfying assumptions) to represent multiple processes. The component of MAAT that is somewhat unique is a system model wrapper. The wrapper is agnostic to the details of the system model, yet can interpret system-model specific input data to set up and run ensembles of models that vary in their process representation, parameter values, and boundary conditions. These ensembles can be set up to perform formal and informal sensitivity analyses of model output with variable model assumptions.

A number of existing modelling codes in the domain of hydrology have similar, multi-assumption capabilities (Downer and Ogden, 2004; Clark et al., 2015; Coon et al., 2016). These different hydrological codes have various purposes and thus different strengths, but are all built to allow flexible model structure within a single overall code structure. The Gridded Surface Subsurface Hydrologic Analysis (GSSHA) code (Downer and Ogden, 2004) is designed for predictive application to specific watersheds. The structural flexibility in GSSHA is primarily intended to allow the tailoring of model structure to suit specific applications and specific watersheds that can differ in their dominant processes. The Structure for Unifying Multiple Modeling Alternatives (SUMMA) (Clark et al., 2015) is designed as a unifying system to organise and compare alternative modelling





approaches. Three main areas of model struture can be altered and compared within SUMMA: 1) alternative modelling domains and their discretisation, 2) alternative process representations, and 3) numerical solutions to the system of process equations across the domain. The Advanced Terrestrial Simulator (ATS) (Coon et al., 2016) is similar to SUMMA but provides an additional capability in that the system model need not be prespecified. ATS has the capacity to build alternative system models,

that differ in complexity, based solely on the particular representation of process that are selected. MAAT complements these other multi-assumption modelling systems by being designed to configure and run large ensembles for process-level sensitivity analyses.

We previously identified process-level sensitivity analysis methods that account for process representation variability were not available and so developed a suitable method (Dai et al., 2017). This sensitivity analysis method is incorporated in MAAT

but is computationally expensive (see Section 3) with a single sensitivity analysis requiring millions of simulations for convergence. For example, a sensitivity analysis of three processes in the photosynthesis model required 100 million simulations, taking five hours on a single computer node of 32 cores. We are pleased to have a 100 million ensemble runtime down to five hours, especially in a scripting language such as R. However, with the current HPC method employed in MAAT we are at the limit of computational scalability. A single instance of the photosynthesis model runs quickly, and models of increased

complexity will require both longer run times for a single ensemble member and more iterations due to larger numbers of processes under investigation (ensemble number is proportional to the number of processes in the analysis). We are currently working to both increase the computational efficiency (reduce the ensemble number) of the sensitivity analysis algorithm, as well as expand the capability of MAAT to operate across multiple compute nodes of an HPC system.

Beven (2006) argues that equifinality in both parameters and process representations is pervasive in models of complex

natural systems and must be embraced by shifting focus from a search for a single optimal model to determining suites of "behavioural" models. Beven (2006) contends that sets of models should be compared against data to determine which models are behavioural depending on certain criteria that scores model output relative to the data, accounting for uncertainty in the data. Models not behavioural should be rejected, while all models that are behavioural should be considered when making predictions about a system. The MAAT modelling system provides a tool to incoporate equifinality in day-to-day modelling

activities. However, work remains to be done to develop tools to facilitate the equifinality approach in MAAT. From a practical standpoint, parameter estimation methods and model selection/hypothesis rejection methods are central to the equfinality thesis and assessment of model structural adequacy (Gupta et al., 2012). Moreover, when multiple process representations are available for a given process, parameters common to more than one represenation can often have different values depending on the particular representation. This difference in values of common parameters is illustrated by the explicitly different labelling

of the $g_1$ parameter in Eqs. 14b, c, and d; and also in the unification of the temperature response curves shown in Fig. 6. MAAT currently does not contain parameter estimation algorithms, nor model or hypothesis rejection algorithms. We plan to include these methods as a priority development. Markov chain Monte Carlo (MCMC) is a powerful Bayesian technique to estimate parameters and that can be used to select models, incorporating multiple sources of uncertainty (e.g. Vrugt et al., 2009; Green, 1995; Beven and Freer, 2001).





More conceptually, MAAT cannot address all elements of epistemic uncertainty in process knowledge and the equifinality thesis. Epistemic uncertainty in process knowledge is necessarily restricted in MAAT to those hypotheses and assumptions that are coded into the modelling system. Alternative hypotheses may exist that have not been discovered by MAAT developers, and MAAT certainly cannot generate hypotheses that may better describe the real-world process or phenomenon than any currently

existing hypothesis. Therefore the full space of epistemic uncertainty can not be explored (Beven, 2016).

Scale and the multiple levels of organisation in biological systems adds a further dimension of complexity. What can be considered a system at one level of organisation can often be represented as a single process at the level of organisition above. For example, the network of interactions that cause an up-regulation of gene transcription in response to an external stimuli to modify a phenotype can often be considered in terms of the environmental stimuli eliciting a phenotypic response without

explicitly modelling the system of genes which effect the change in phenotype. Different levels of complexity in the system model itself is also worth noting, e.g. enzyme kinetic vs light use efficiency, or energy balance and representation of leaf boundary layer. This is dealt with in MAAT by specifying the overarching system model as a variable assumption and allows the rapid development of alternative conceptualisations of the system as a whole.

Additional work and conceptual limitations notwithstanding, MAAT is a powerful new tool that can be used to understand

the sensitivities of photosynthesis to variation in assumptions and mechanistic hypotheses made to represent photosynthetic processes. More broadly, the agnosticism of the wrapper allows rapid incorporation of new assumptions and development of new system models, without any overhead in development of the wrapper. This model-system agnostic wrapper forms the core of MAAT and over time we hope it will be used to facilitate the development and analysis of models in many different scientific domains. Once a few simple rules are learned on how to write a system model in the MAAT formalism, MAAT provides an

ideal testbed for novel model development and for developing stand-alone components of more complex models, allowing a full analysis of internal model dynamics and response to boundary conditions. Should researchers wish to develop system models, 'toy' models, and stand-alone components of larger models we encourage them to download the code and resources.

## 7   Summary

The MAAT modelling system embraces the equifinality thesis, "the potential for multiple acceptable models as representations

of hydrological and other environmental systems" (Beven, 2006). We also contend that no matter which side of the debate one tends to sit (the quest for a single optimal model vs use of suites of behavioural models) there are currently, and most likely will be for many years to come, many different models used to simulate almost any given system. So long as this multiplicity is the norm we need better tools to understand the causes of differences among models and to understand the consequences of adding new processes or different process representations to a model. The multi-assumption architecture and testbed has

been developed as a tool to facilitate and formalise this approach to modelling. We hope that MAAT and other tools like it will enable researchers in the environmental sciences to gain a deeper and more quantitative understanding of their study system.



# 8 Code availability

Code is available from https://bitbucket.org/walkeranthonyp/maat .

# 9 Data availability

The limited data used in this publication will be made available on the repository as part of code tutorials.

*Author contributions.* APW conceived of and wrote MAAT, wrote the paper and ran the analysis. MY and DL provided code to implement the sensitivity analysis ensembles and calculate the sensitivity indices. MDK provided guidance on object oriented programming. MDK, LG, BM, AR, and SS all provided feedback on the development of the unified multi-assumption model of leaf-scale C3 photosynthesis. All authors provided feedback on the manuscript during drafting.

*Competing interests.* None.

*Acknowledgements.* The MAAT modelling framework and sensitivity analysis component of this research was supported as part of the ORNL Terrestrial Ecosystem Science, Science Focus Area, funded by the U.S. Department of Energy, Office of Science, Office of Biological and Environmental Research. The multi-assumption leaf-scale photosynthesis model component of this research was supported as part of the Next Generation Ecosystem Experiments-Tropics, funded by the U.S. Department of Energy, Office of Science, Office of Biological and Environmental Research. Oak Ridge National Laboratory is operated by UT-Battelle, LLC, under contract DE-AC05-00OR22725 to the

United States Department of Energy. Brookhaven National Laboratory is managed under contract No. DE-SC0012704 to the United States Department of Energy. We thank Lisa Jansson (BNL) for assistance with graphic design.



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
