# Peer review of "The Multi-Assumption Architecture and Testbed (MAAT v1.0): R code for generating ensembles with dynamic model structure and analysis of epistemic uncertainty from multiple sources"

_Geoscientific Model Development, 2018_

## Referee Comment (RC1) · N. Smith (Referee) · 10 May 2018

The paper "The Multi-Assumption Architecture and Testbed (MAAT v1.0): Code for ensembles with dynamic model structure including a unified model of leaf-scale C3 photosynthesis" by Walker et al. describes a newly developed testbed for assessing parameter and structural uncertainty in mathematical models. The paper introduces and describes the MAAT system. The paper also tests the parameter sensitivity analysis component of the system using a comparison with a previously published paper utilizing a simple groundwater model. Finally, the paper examines the formulation un-

certainty component of the system using a leaf-scale photosynthesis model. In each case, MAAT performs as intended, demonstrating a great deal of potential benefit to researchers working with models of systems (biological, geological, and beyond).

My largest criticism, and it is minor, is that the broad implementation of MAAT is, as presented, somewhat difficult to envision. The authors note that "once a few simple rules are learned on how to write a system model in the MAAT formalism, MAAT provides an ideal testbed for novel model development and for developing stand-alone components of mode complex models..." However, I'm left wondering what these few simple rules are and whether there is an explicit protocol for integrating different model systems into the MAAT framework. I was not able to find this on the bitbucket site. This would substantially broaden the impact of the paper. While I personally find the integrated plant physiology model useful, the reach of the MAAT system could be exponentially greater if used by other communities (as is noted by the authors). A brief section on integration of different models into MAAT would be useful.

Related to the point above, there is a lot of text devoted to describing the photosynthesis models. While this is great information, it may be a bit distracting to readers that are not interested in the plant physiology responses per se, but rather the capabilities of the MAAT system. Many of the details could be included as supplement. This is not critical, but would improve readability.

Smaller concerns:

Title: The title does not address the model's capacity to evaluate multiple sources of epistemic uncertainty, which seems to be the best feature! Also, I think the connection to the photosynthesis model does not necessarily need to be in the title and may limit the reach of the paper.

P1L8: More completely than what?

P5L5: Cite the proto package. citation('proto')

[Figure]

Figure 3: I'd suggest increasing the font size on axes and axis titles.

P18L2: Should this be gs = (1/rs) ?

P20L4: You could just say "a form." It seems unnecessary to speculate whether it's the most common form.

P26L8: It is my understanding that the Tjoelker et al. (2001) Q10 respiration is not acclimation per se, but just a modification of the instantaneous response that allows for the observed dampening of the exponential slope with increasing leaf (not acclimated) temperature

Figures 4 and 5: Check that the axis text does not overlap.

Figure 4 legend: "analystical" should be "analytical"

---

## Referee Comment (RC2) · Anonymous Referee #2 · 10 May 2018

The manuscripts presents MAAT, an R-based interface to assess epistemic uncertainty and its sources within and between models. The tool is validated using a simple groundwater model and an application of MAAT is presented using different process representations for the C3 pathway of photosynthesis and stomatal conductance, key components in all land surface schemes for ecosystem and climate models. A modelling tool that serves as a testbed for such an uncertainty analysis is definitely useful. However I have a few comments and criticisms for the present manuscript:

- As far as I understand, MAAT needs all modelling components to be written or

wrapped in R functions (Please correct if I am wrong and clarify in the manuscript). I am a bit worried that this will need substantial recoding, especially for models written in different languages (e.g. compiled code from C, FORTRAN can be called through R, but what about interpreter languages such as e.g. Matlab). This kind of recoding might be unfeasible if someone needs to perform uncertainty analysis for a specific component of a large model (e.g. land surface model [e.g. CLM], dynamic vegetation models [e.g. ORCHIDEE, LPJ, ED etc.]) that involves several thousand lines of code. Can the authors give more detail on the applicability of their tool? I might be wrong, but it is worth clarifying the limits of applicability of MAAT.

-The scope of the manuscript is to present MAAT. However much more detail is given on the description of the application (C3 photosynthesis, stomatal conductance models). I would expect more detail on the algorithms of MAAT. Details of the models of photosynthesis and stomatal conductance can be presented in a supplementary since anyhow have been presented elsewhere.

-Linking to my previous comment, lines 3-20 in page 10 that describe the key algorithms in MAAT need to be presented more rigorously. A better explanation of the matrices A, B and AB(i) is also needed.

-Since MAAT is a testbed for uncertainty analysis, I would expect a number of uncertainty/sensitivity metrics, similar to the ones presented in Table 2, also for the detailed photosynthesis application. In the present manuscript the authors state that "the purpose of the simulations is to verify the photosynthesis code", but since this is an application of MAAT it is worth actually presenting the uncertainty/sensitivity results that MAAT can produce. Validation of the photosynthesis functions could be moved to a supplementary file, since I believe this is not the focus of the paper. The authors might want to consider restructuring their results accordingly.

-In several points throughout the manuscript the authors claim that epistemic uncertainty linked to process representation between models has not been treated formally

in the past. There is a big exception in that in climate science related to climate models (see some references), where the literature is vast, especially when it comes to multi-model ensembles. I believe this is worth a discussion point.

References:

Knutti, R., & Sedláček, J. (2013). Robustness and uncertainties in the new CMIP5 climate model projections. Nature Climate Change, 3(4), 369.

Tebaldi, Claudia, and Reto Knutti. "The use of the multi-model ensemble in probabilistic climate projections." Philosophical Transactions of the Royal Society of London A: Mathematical, Physical and Engineering Sciences 365, no. 1857 (2007): 2053-2075.

Knutti, Reto, Reinhard Furrer, Claudia Tebaldi, Jan Cermak, and Gerald A. Meehl. "Challenges in combining projections from multiple climate models." Journal of Climate 23, no. 10 (2010): 2739-2758.

---

## Referee Comment (RC3) · M. Cailleret (Referee) · 11 May 2018

The paper describes a new modeling framework developed under R that can be used to estimate the epistemic uncertainty in any system model. This new tool was tested on a leaf-scale photosynthesis model whose internal processes and state variables can be simulated following several approaches or equations. Such tool would be definitely useful for the modeling community irrespective of their system of interest (global vegetation, hydrological cycles, forests etc.); however the present manuscript has some minor issues, and should be clarified in some aspects:

[Figure]

First, there is a problem with the structure of the paper. (1) Relatively to the description of MAAT, the photosynthesis model is highly detailed. This is unbalanced, especially when looking at the abstract where only 3 sentences are devoted to the photosynthesis model. (2) The section P2-L31 to P3-L12 looks like M&M. At least I would not include it in the Introduction. (3) I would move the section 2.4 (HPC) at the end of the section 2.1. During my first reading of P6, I was wondering why the classical 'apply' functions were used (L16), while functions from the parallel R package were not... This information should be better located here. (4) I'm wondering if the section 4.3 could be better placed at the end of the section 4: the Brent solver can be used for any other equations/models that have to be solved, not only to calculate Cc..

Second, the photosynthesis model simulates many different processes. For some of them, only one function (∼equation) can be used (e.g., Cc ; eq. 13c) , while for others, there are many possibilities(e.g., photorespiratory compensation point; gs; etc.). It would be great for the reader to include an additional table, which summarizes the processes for which many functions can be used (e.g., processes as rows; equation number as columns); it would also highlight the high number of combinations generated by the factorial simulation design.

Finally, I'm quite frustrated to not see any sensitivity statistics in the section 5 and in the discussion. It may be intended for another paper, but the reader of the present paper can feel disappointed to not see any result (e.g., what is the uncertainty in A due to the selection of the stomatal model? see Figure 5).

Other comments:

It may worth mentioning in the title and/or in the abstract that this modeling framework was developed under R

P2-L3: "unforeseen pattern". Do you mean emerging pattern sensus Levin (1992; Ecology)?

[Figure]

P2-L24: "ensemble of possibilities" rather than "ensemble of opportunity"?

P2-L24: add MIP after inter-comparison projects

P2-L25: "models in the ensemble are not independent of another. Correct, here I would give as example the genealogy of global climate models developed by Masson & Knutti 2011 Geo. Res. Lett.

P4-L7: As method to estimate model sensitivity to variable process representation, I would mention the classical variance partitioning approaches ($\sim$uncertainty decomposition; e.g., Nishina et al. 2015 Earth Syst. Dynam.; or better Dietze 2017 Ecol. Appl.)

P6-L1 to L4: It is not very clear what you mean by 'run', 'run function', 'run script', 'runtime'; e.g., what is the difference between a script and a 'run script'? Because of that, I misunderstood what you meant by "can configure the ensemble of all possible combinations of these choices during run time" (P2-L34), and "allowing variable assignment during runtime" (P1-L9; P7-L1). I initially thought that you could change functions to simulate a given process within a run of your photosynthesis model (e.g., for simulating gs: Ball et al. 1987 function at time step 1, Leuning (1990) function at time step 2, Medlyn function at time step 3, Leuning at time step 4 etc.), which would have been strange, and not adequate to calculate sensitivity indices. Rather, your script generates a complete factorial combination of options, and then run the photosynthesis model for each combination of parameters/variables/functions (P9). I would remove these 'during runtime' statements. I would also use the term 'run' only when you do (run) a simulation of your photosynthesis model (which could be an hydrological model, a DGVM, etc.).

P6-L27 to L29: there is a contradiction in your definition of 'secondary state variables'. On one side, you write that they 'can be thought as dynamic parameters' (L17), but on the other side, they 'are fixed parameters 'L29). Please clarify.

P6-L30: I don't really understand the sentence... could you rephrase?

[Figure]

Figure 2: "Panel a) represents the first two steps". Not really, the step 3 'run ensemble' is also represented. Also, I'm wondering if the color information of the arrows (call, write, read) is really needed: this is a technical aspect, which makes the figure more difficult to understand.

P9-L22: remove 'ensembles"

P23-L12 to L13: any word missing here?

P24-L19 to L21: not clear, please explicitly mention that optimums are reached at process-specific thresholds after which rates decrease.

P25-L17: add "after an optimal value" after "with higher temperatures"

---

## Author Comment (AC1) · 13 Jul 2018

**Response to Referees:** manuscript gmd-2018-71

Revised title:

The Multi-Assumption Architecture and Testbed (MAAT v1.0): R code for generating ensembles with dynamic model structure and analysis of epistemic uncertainty from multiple sources

We appreciate the supportive and constructive nature of the reviews, and we thank the reviewers for their time and thoughtful comments. We have carefully considered each of the reviewers' comments and incorporated the majority of them in the revised manuscript which, in our view, has improved the clarity and structure of the manuscript. Please see below our detailed responses (added in blue for ease of reading, with direct quotations from the revised manuscript in bold) to each of the points made by each of the reviewers.

Please note the we have moved the repo to GitHub for more visibility:

https://github.com/walkeranthonyp/MAAT

Reviewer 1.

The paper "The Multi-Assumption Architecture and Testbed (MAAT v1.0): Code for ensembles with dynamic model structure including a unified model of leaf-scale C3 photosynthesis" by Walker et al. describes a newly developed testbed for assessing parameter and structural uncertainty in mathematical models. The paper introduces and describes the MAAT system. The paper also tests the parameter sensitivity analysis component of the system using a comparison with a previously published paper utilizing a simple groundwater model. Finally, the paper examines the formulation uncertainty component of the system using a leaf-scale photosynthesis model. In each case, MAAT performs as intended, demonstrating a great deal of potential benefit to researchers working with models of systems (biological, geological, and beyond).

We thank the reviewer for their support and are happy that the reviewer sees the potential in our work.

My largest criticism, and it is minor, is that the broad implementation of MAAT is, as presented, somewhat difficult to envision. The authors note that "once a few simple rules are learned on how to write a system model in the MAAT formalism, MAAT provides an ideal testbed for novel model development and for developing stand-alone components of mode complex models. . ." However, I'm left wondering what these few simple rules are and whether there is an explicit protocol for integrating different model systems into the MAAT framework. I was not able to find this on the bitbucket site. This would substantially broaden the impact of the paper. While I personally find the integrated plant physiology model useful, the reach of the MAAT system could be exponentially greater if used by other communities (as is noted by the authors). A brief section on integration of different models into MAAT would be useful.

This is a fair comment and we had not yet found the time to describe the MAAT formalism in detail. We have added substantial description to the MAAT repo in a README in the src/system_models directory. Please note the we have moved the repo to GitHub for more visibility (https://github.com/walkeranthonyp/MAAT).

We would prefer to not be too specific in how to integrate different models in MAAT in the manuscript itself as the details are prone to change over time and these details would be better suited to the living documents housed in the repo. We have added a paragraph to Section 2 (the MAAT description section) of the manuscript that points to these READMEs.

**The MAAT source code is available on GitHub (https://github.com/walkeranthonyp/MAAT) and READMEs that come with the source code provide: guidance on how to set up and run MAAT; some examples of using MAAT to generate the data and some of the figures presented in this paper; and details of the MAAT formalism and how to code a new model object. How to develop a new system model in MAAT is detailed in these READMEs as well as how to integrate new process representations in an existing system model. We recommend starting with the README in the highest level directory of the source code as this provides the very initial guidance needed to set up MAAT and points to the other READMEs for more advanced information.**

Please also see our comment and text added to the manuscript in response to a comment on model integration by reviewer 2.

Related to the point above, there is a lot of text devoted to describing the photosynthesis models. While this is great information, it may be a bit distracting to readers that are not interested in the plant physiology responses per se, but rather the capabilities of the MAAT system. Many of the details could be included as supplement. This is not critical, but would improve readability.

This point was one of discussion when pulling the manuscript together. On the one hand we wanted to provide a resource that presents multiple diverse photosynthesis models with multiple notations in the literature into a single unified notation and representation. However, the photosynthesis model description does somewhat overwhelm the other, more general component of the manuscript and MAAT. We have moved the majority of the photosynthesis model description to an appendix. To the main text we add a table describing the various processes and their multiple representations as suggested by Reviewer 3. The text in the main document outlining the photosynthesis model now reads:

**Photosynthesis is a central process of the biosphere. At the heart of many Terrestrial Ecosystem and Biosphere Models (TBMs) lie the mathematical hypotheses describing the enzyme kinetics of photosynthesis and the hypotheses and assumptions describing associated processes, e.g. stomatal conductance. Enzyme kinetic models lie at the core of TBMs in order to accurately simulate the ecophysiological interaction of terrestrial ecosystems with the interrelated carbon, water, and energy cycles of the Earth System. Many studies have demonstrated the sensitivity of TBM predictions to variation in parameters and assumptions used to represent these core model processes (e.g. Zaehle et al., 2005; Rogers et al., 2017; Anav et al., 2015; Bonan et al., 2011; Walker et al., 2017b).**

**In Appendix A we describe in detail the unified, multi-assumption model of leaf-scale photosynthesis. The current focus is on enzyme kinetic models of photosynthesis (Farquhar et al., 1980; von Caemmerer, 2000) rather than light use efficiency models. Enzyme kinetic and light use efficiency models can be thought of as alternative conceptualisations of the leaf photosynthesis system. Enzyme kinetic models were the first photosynthesis conceptualisation to be built into MAAT as they are the most commonly employed photosynthesis model employed by TBMs. Alternative representations for individual processes are listed in Table 4.**

**In this section we present the results from some simulations with MAAT. The purpose of these simulations is to verify that the photosynthesis code is working as intended, not to test various implementations against data which we will save for extensive evaluations in future research. The use of both numerical and analytical solutions to the system of simulataneous equations for photosynthesis, as well as multiple instances of stomatal conductance equations (with some designed for analytical solution), provides a testbed for code verification. We also demonstrate a simple comparison among the temperature response functions.**

Smaller concerns:

Title: The title does not address the model's capacity to evaluate multiple sources of epistemic uncertainty, which seems to be the best feature! Also, I think the connection to the photosynthesis model does not necessarily need to be in the title and may limit the reach of the paper.

We have revised the title as follows:

**The Multi-Assumption Architecture and Testbed (MAAT v1.0): R code for generating ensembles with dynamic model structure and analysis of epistemic uncertainty from multiple sources**

P1L8: More completely than what?

deleted "more"

P5L5: Cite the proto package. citation('proto')

There is no citation for the proto package. We now cite the GitHub repo for the proto package.

Figure 3: I'd suggest increasing the font size on axes and axis titles.

Done.

P18L2: Should this be gs = (1/rs) ?

This was confusing, we have changed to $g_s$ ($g_s=1/rs$).

P20L4: You could just say "a form." It seems unnecessary to speculate whether it's the most common form.

Have changed to **A form of … commonly used by TBMs.**

P26L8: It is my understanding that the Tjoelker et al. (2001) Q10 respiration is not acclimation per se, but just a modification of the instantaneous response that allows for the observed dampening of the exponential slope with increasing leaf (not acclimated) temperature.

Good point, thanks for catching. We had had some discussion about this during ms preparation but the lead author obviously did a bad job of following up on that! We have moved the Tjoelker representation to the instantaneous temperature response section and modified the text as follows:

**Tjoelker et al. (2001) demonstrated that the logarithm of respiration plotted against measurement temperature was not a linear function. The inference was made that Q 10 was a function of measurement temperature. This is somewhat confusing as the Q 10 function describes the response to temperature. Our interpretation of the evidence presented in Tjoelker et al. (2001) is that the R d temperature response was not a true exponential function and therefore a Q 10 function is not the correct representation of the R d temperature response. We include the Tjoelker et al. (2001) function that describes the parameter Q 10 as a function of leaf temperature for completeness as it is used in some TBMs.**

Figures 4 and 5: Check that the axis text does not overlap.

Done.

Figure 4 legend: "analystical" should be "analytical"

Thanks, done.

Reviewer 2.

The manuscripts presents MAAT, an R-based interface to assess epistemic uncertainty and its sources within and between models. The tool is validated using a simple groundwater model and an application of MAAT is presented using different process representations for the C3 pathway of photosynthesis and stomatal conductance, key components in all land surface schemes for ecosystem and climate models. A modelling tool that serves as a testbed for such an uncertainty analysis is definitely useful.

We are happy that the reviewer sees the utility in this work.

However I have a few comments and criticisms for the present manuscript:

- As far as I understand, MAAT needs all modelling components to be written or wrapped in R functions (Please correct if I am wrong and clarify in the manuscript). I am a bit worried that this will need substantial recoding, especially for models written in different languages (e.g. compiled code from C, FORTRAN can be called through R, but what about interpreter languages such as e.g. Matlab). This kind of recoding might be unfeasible if someone needs to perform uncertainty analysis for a specific component of a large model (e.g. land surface model [e.g. CLM], dynamic vegetation models [e.g. ORCHIDEE, LPJ, ED etc.]) that involves several thousand lines of code. Can the authors give more detail on the applicability of their tool? I might be wrong, but it is worth clarifying the limits of applicability of MAAT.

The reviewer is correct that all modelling functions must be written in R for MAAT to use them. A small point for clarity re the reviewers first comment: MAAT is not really an interface, it is a stand alone piece of software, to make this clear we add the following line to the final paragraph of the introduction:

**The main components of MAAT are a software wrapper to generate and run the ensemble, an interface to pass assumptions to a system model, and a system model. All of these components are coded in R.**

Currently, there is no interface with C or FORTRAN code, nor Matlab, etc. Though we are investigating how to interface with C and FORTRAN. However, even if MAAT could call existing model code, existing code is very often nowhere near sufficiently modular to be able to pose all possible models or break the sensitivity analysis down by process. This level of modularity is necessary to properly analyse process representation uncertainty (as represented by the method we developed in Dai et al., 2017). Therefore existing code very often (in our experience in the vast majority of cases) is not suitable for formal and correct analysis of variation in model output caused by process representation. To do this would require substantial recoding of existing code. This was one of the main reasons for developing MAAT. However, in many cases, we argue the time invested in recoding is scientifically worthwhile. New models and modelling architectures are being developed all the time and we argue that this agile and flexible software development is the way of the future.

Notably even the photosynthesis code in large terrestrial biosphere models is several thousand lines long. CTSM/E3SM leaf photosynthesis code is in the region of 2000 lines. In MAAT the photosynthesis code is about the same length (1829 lines) and can be applied to mimic almost any model of the CMIP ensemble, and pose every single possible combination thereof.

We agree with the reviewer that there are limitations to MAAT, especially as it currently can only be applied to photosynthesis code. Eventually we envision an ecosystem scale model. This will take time to code of course, but the value will be that novel conceptualisations of processes and hypotheses will be very simple to incorporate in the MAAT framework and examine in the systems context.

We add the following paragraph to the discussion:

**An additional practical limitation of MAAT is that models must be coded in R in the MAAT formalism, which comes at a cost. Currently, there is no interface for MAAT to interact with existing model code though we are investigating a possible C and FORTRAN interface. However, even if MAAT could call existing model code, very often existing code is nowhere near sufficiently modular to extract individual process representations. This level of modularity is necessary to fully explore process representation uncertainty, thus existing code very often (in our experience in the vast majority of cases) would require substantial recoding to acheive the required level of modularity. We suggest that in many cases, the time invested in recoding models into R in the MAAT formalism is scientifically worthwhile. Once a system model has been coded in MAAT, novel conceptualisations of processes and hypotheses are very simple to incorporate and examine in the systems context. New models and modelling architectures are being developed all the time and we argue that this agile and flexible style of software development will help to rapidly and robustly develop and assess new process representations. Currently MAAT can only be applied to photosynthesis code, which runs relatively rapidly and requires no spin-up of state-variables. Eventually we envision an ecosystem scale model coded within MAAT. An ecosystem scale model with many, many processes and requiring spin-up of state variables will increase model runtime and MAAT may need to interface with compiled languages to maximise computational efficiency.**

```
-The scope of the manuscript is to present MAAT. However much more detail is given
on the description of the application (C3 photosynthesis, stomatal conductance
models). I would expect more detail on the algorithms of MAAT. Details of the
models of photosynthesis and stomatal conductance can be presented in a
supplementary since anyhow have been presented elsewhere.
```

We have moved the photosynthesis model descriptions to an appendix, see additional detail in the response to Reviewer 1. This weights the description much more in terms of the MAAT algorithms than the unified photosynthesis model.

```
-Linking to my previous comment, lines 3-20 in page 10 that describe the key
algorithms in MAAT need to be presented more rigorously. A better explanation of
the matrices A, B and AB(i) is also needed.
```

We agree this text was a little short on detail. We have edited the manuscript as follows please excuse the small errors in rendering, we made the edits in the ms in latex while compiling this document in word:

**The algorithms for the parameter and the process sensitivity indices are not simply factorial combinations of process representations and parameters (Dai et al., 2017). Therefore the configuration of the 'fnames' and 'pars' matrices and the run cascade is different for each of the algorithms. The algorithms are described in detail in Saltelli et al. (2010) and Dai et al. (2017) so we do not go into great detail here.**

For the parameter sensitivity algorithm (Jansen, 1999; Saltelli et al., 2010), two parameter sample matrices are constructed, A and B, both with n rows and $n_p$ columns, where n and $n_p$ are the number of samples and the number of parameters in the sensitivity analysis. Each row of these matrices contains a sample from the distributions of each parameter (columns) in the analysis. A further $n_p$ parameter matrices, $A_B^{(i)}$, are constructed by copying the A matrix and replacing the parameter samples in column i of matrix $A_B^{(i)}$ with column i from the B matrix. For a single model, the model is run once for each row of the $2 + n_p$ parameter sample matrices ( A, B, and $A_B^{(i)}$) using the parameter values in the row. The first order, $S_i$, and total sensitivity, $S_{Ti}$, indices are calculated after Jansen (1999), see Table 2 (Saltelli et al., 2010):

$$S_i = \frac{V\{Y\} - \frac{1}{2n} \sum_{j=1}^{n} (f(\mathbf{B}_j) - f(\mathbf{A_B^{(i)}}_j))^2}{V\{Y\}}$$

$$S_{Ti} = \frac{\frac{1}{2n} \sum_{j=1}^{n} (f(\mathbf{A})_j - f(\mathbf{A_B^{(i)}})_j)^2}{V\{Y\}}$$

where V {} is the variance function, f() is the model, and Y = f (A, B) is the model output when evaluated across all rows of matrices A and B.
When multiple models are available, the parameter sensitvity indices are calculated for each model combination. Each model combination is run over matrices A, B, and $A_B^{(i)}$. As MAAT is designed to switch in alternative assumptions (hypotheses, representations, or structures) for each process in the analysis, the number of all possible models is $\prod_{k=1}^{n_k} \phi_k$ , where $n_k$ is the number of processes in the sensitivity analysis and $\phi_k$ is the number of representations of process k. With both variable processes and parameters, the total number of individual model runs in this algorithm is: $(2+n_p)n\prod_{k=1}^{n_k} \phi_k$ .

The process sensitivity algorithm (Dai et al., 2017) is a set of five nested loops. The outer (first) loop iterates over each of the $n_k$ processes in the sensitivity analysis. The second loop iterates over each of the $\phi_k$ representations of process k. The third loop iterates over a parameter matrix $A^{(k)}$ of n rows and $n_{pk}$ columns where n is the number of samples and $n_{pk}$ is the number of parameters in process k. The fourth loop iterates over the factorial combination of the $\phi_{\sim k}$ representations of all the other processes in the analysis. The fifth (inner) loop iterates over parameter matrix $A^{(\sim k)}$ of n rows and $n_{p \sim k}$ columns, where $n_{p \sim k}$ is the number of parameters in all other processes $\sim k$. The total number of iterations in the process senstivity analysis is: $n_k n^2 \prod_{k=1}^{n_k} \phi_k$ . The function to evaluate the first order process sensitivity index is as follows (Dai et al., 2017):

$S_k = V\{Y\}_k / V\{Y\}$

where Y is the array of model output evaluated across all model combinations and parameter samples; and $V\{Y\}_k$ is the partial variance in model output caused by variation in process k:

$$V\{Y\}_k = \sum_{l=1}^{\phi_k} P_{k,l} \left( EE_{k,l} - E_{k,l}^2 \right)$$

**where $P_{k,l}$ is the probability of representation l of process k (assumed equal across all representations), and:**

$$EE_{k,l} = \frac{1}{n} \sum_{j=1}^{n} E_{k,l,j}{}^2$$

$$E_{k,l} = \frac{1}{n} \sum_{j=1}^{n} E_{k,l,j}$$

**and:**

$$E_{k,l,j} = \frac{1}{n} \sum_{m=1}^{\Pi \phi_{\sim k}} P_{\sim k,m} \sum_{o=1}^{n} f_{k,l} f_{\sim k,m}\left(\mathbf{A}^{(\mathbf{k})}{}_{\mathbf{j}}, \mathbf{A}^{(\sim \mathbf{k})}{}_{\mathbf{o}}\right)$$

**where $E_{k,l,j}$ is an array of model output averaged across dimension o (parameter samples from matrix A(~k)). $f_{k,l}$ $f_{\sim k,m}(\mathbf{A}^{(\mathbf{k})}{}_{\mathbf{j}}, \mathbf{A}^{(\sim \mathbf{k})}{}_{\mathbf{o}})$ represents a single model run using representation l of process k and the combination of representations m of processes ~k evaluated with the parameter samples $A^{(\mathbf{k})}{}_{j}$ and $A^{(\sim \mathbf{k})}{}_{o}$ . $P_{\sim k,m}$ is the probability of the combination of representation m of process ~k (assumed equal across all combinations).**

-Since MAAT is a testbed for uncertainty analysis, I would expect a number of
uncertainty/sensitivity metrics, similar to the ones presented in Table 2, also for
the detailed photosynthesis application. In the present manuscript the authors
state that "the purpose of the simulations is to verify the photosynthesis code",
but since this is an application of MAAT it is worth actually presenting the
uncertainty/sensitivity results that MAAT can produce. Validation of the
photosynthesis functions could be moved to a supplementary file, since I believe
this is not the focus of the paper. The authors might
want to consider restructuring their results accordingly.

MAAT is a testbed for uncertainty analysis, and we verify that this code is working correctly with the ground water test case. Our intention was not an application of MAAT. We show results from the photosynthesis code to verify that this code is working correctly. This is in accordance with the "model description" type of manuscript described by GMD (https://www.geoscientific-model-development.net/about/manuscript_types.html#item1) and our intended type for this manuscript. We do not perform a sensitivity analysis of the photosynthesis code as this is not within our intended scope for the manuscript, our intended scope (as pointed out by the reviewer in a previous comment) is to present MAAT, describe MAAT and the unified photosynthesis code, and verify that the code is working correctly as a solid foundation and reference for future research.

-In several points throughout the manuscript the authors claim that epistemic
uncertainty linked to process representation between models has not been treated
formally in the past. There is a big exception in that in climate science related
to climate models (see some references), where the literature is vast, especially
when it comes to multi-model ensembles. I believe this is worth a discussion point.

This is a very fair comment, and we agree with the reviewer. It was not our intention to lump this vast literature in with informal treatment of process representation epistemic uncertainty. Would it be fair to

say that the formal treatment of epistemic uncertainty the reviewer refers to is post-hoc, and somewhat incomplete given the small subset of possible models contained within any ensemble?

We now qualify our statement in the Abstract:

**Many formal methods exist to analyse parameter-based epistemic uncertainty, while process-representation based epistemic uncertainty is often analysed post-hoc, incompletely, informally, or is ignored.**

And add a discussion of these formal methods to the Introduction:

**Often process representation uncertainty is assessed by analysing the cross-model variability in the ensembles of model intercomparison projects (MIPs) (Refsgaard et al., 2007; Friedlingstein et al., 2014; Herger et al., 2018). These ensembles can be thought of as ensembles of opportunity and capability (Tebaldi and Knutti, 2007), the ensemble members are determined by the opportunity and the capability of the modelling teams to contribute results. A large body of literature has developed and employed formal statistical techniques for post-hoc analysis of these ensembles of opportunity (e.g., Refsgaard et al., 2006; Herger et al., 2018; Knutti et al., 2009). These formal analyses account for non-independence of models in the ensemble (e.g., Masson and Knutti, 2011), can weight models based on how well they reproduce observed data (e.g., Fang and Li, 2015), and subset the ensemble for improved performance and reduced unsertainty (e.g., Herger et al., 2018); yielding a more robust estimate of the process representation uncertainty of the ensemble. However, these ensembles do not represent an a priori assessment of process representation uncertainty. A full a priori assessment of process representation uncertainty involving clear delination of which representations to employ for each modelled process and a factorial combination of these options to create an ensemble of all possible models is rarely, if ever, done.**

References:
Knutti, R., & Sedláček, J. (2013). Robustness and uncertainties in the new CMIP5 climate model projections. Nature Climate Change, 3(4), 369.

Tebaldi, Claudia, and Reto Knutti. "The use of the multi-model ensemble in probabilistic climate projections." Philosophical Transactions of the Royal Society of London A: Mathematical, Physical and Engineering Sciences 365, no. 1857 (2007): 2053-2075.

Knutti, Reto, Reinhard Furrer, Claudia Tebaldi, Jan Cermak, and Gerald A. Meehl. "Challenges in combining projections from multiple climate models." Journal of Climate 23, no. 10 (2010): 2739-2758.

Reviewer 3.

The paper describes a new modeling framework developed under R that can be used to estimate the epistemic uncertainty in any system model. This new tool was tested on a leaf-scale photosynthesis model whose internal processes and state variables can be simulated following several approaches or equations. Such tool would be definitely useful for the modeling community irrespective of their system of interest (global vegetation, hydrological cycles, forests etc.); however the present manuscript has some minor issues, and should be clarified in some aspects:

We are happy that the reviewer sees the utility in this work.

First, there is a problem with the structure of the paper. (1) Relatively to the description of MAAT, the photosynthesis model is highly detailed. This is unbalanced, especially when looking at the abstract where only 3 sentences are devoted to the photosynthesis model.

We have moved the description of the photosynthesis model to an appendix. Please see response to Reviewer 1 for more detail.

(2) The section P2-L31 to P3-L12 looks like M&M. At least I would not include it in the Introduction.

We agree that these paragraphs were out of place. The justification for the photosynthesis model has been moved to the photosynthesis model section in the main text. We cut some of the MAAT description and move some of it to the final paragraph of the introduction, which now reads:

**In this study we build on previous efforts and present a modular modelling code designed explicitly to be system model agnostic and for the generation of large model ensembles that differ in how each process within a system is represented. We describe the multi-assumption architecture and testbed (MAAT v1.0) is a modelling framework that can formally, systematically, and rigorously analyse variability in system model output caused by variability in process representation, as well as parameters and boundary conditions. MAAT allows users to specify multiple process representations for multiple processes and can configure the ensemble of all possible combinations of these choices during a single execution. The main components of MAAT are a software wrapper to generate and run the ensemble, an interface to pass assumptions to a system model, and a system model. The system model is highly modular by design, allowing flexible model structure according to information passed from the interface. Algorithms to analyse the sensitivity of model outputs to variation in process representations and parameters are contained within the wrapper. While the ensemble generation code is system model agnostic, allowing the analysis of any system model coded in the MAAT formalism, our primary domain of research is biogeosciences and ecosystem ecology. Therefore MAAT v1.0 comes packaged with a unified multi-assumption leaf-scale photosynthesis model as its primary system model.**

(3) I would move the section 2.4 (HPC) at the end of the section 2.1. During my
first reading of P6, I was wondering why the classical 'apply' functions were used
(L16), while functions from the parallel R package were not... This information
should be better located here.

We have moved the HPC section to be a subsection of section 2.1.

(4) I'm wondering if the section 4.3 could be better placed at the end of the
section 4: the Brent solver can be used for any other equations/models that have to
be solved, not only to calculate Cc.

It is true that the Brent solver is a general univariate root finder, but section 4.3 (now A3) describes
how the system of photosynthesis equations can be solved both analytically and numerically. Section
4.1 (A1) describes the carbon assimilation equations and then section 4.2 (A2) resistance to and a
diffusion of $CO_2$ to the site of carboxylation. In or view it is logical to then follow these two sections,
which describe the full set of simultaneous equations, with a section on how those equations are solved.
The following sections 4.4 and 4.5 (A4 and A5) describe how parameters within the equations are
determined and are outside of the solution of the photosynthesis equations. We would prefer to keep the
order as it is.

Second, the photosynthesis model simulates many different processes. For some of
them, only one function (~equation) can be used (e.g., Cc ; eq. 13c) , while for
others, there are many possibilities(e.g., photorespiratory compensation point; gs;
etc.). It would be great for the reader to include an additional table, which
summarizes the processes for which many functions can be used (e.g., processes as
rows; equation number as columns); it would also highlight the high number of
combinations generated by the factorial simulation design.

Great point, we have added a table (Table 4) as described by the reviewer.

Finally, I'm quite frustrated to not see any sensitivity statistics in the section
5 and in the discussion. It may be intended for another paper, but the reader of
the present paper can feel disappointed to not see any result (e.g., what is the
uncertainty in A due to the selection of the stomatal model? see Figure 5).

We understand the reviewers frustration, but is was never our intention to present a sensitivity analysis
of the photosynthesis model. In accordance with the "model description" style of manuscript described
by GMD (https://www.geoscientific-model-development.net/about/manuscript_types.html#item1), and
our intended style for this manuscript, we verify that the sensitivity analysis and photosynthesis code is
working correctly. Moving the photosynthesis code to an appendix, as suggested by all three reviewers,
refocuses the manuscript back towards the description of the MAAT model and its general application
rather than a specific application.

Other comments:

It may worth mentioning in the title and/or in the abstract that this modeling
framework was developed under R

We have added R to the title.

P2-L3: "unforeseen pattern". Do you mean emerging pattern sensus Levin (1992; Ecology)?

This text was a little confusing, the phrase now reads "**… can exhibit complex behaviour.**"

P2-L24: "ensemble of possibilities" rather than "ensemble of opportunity"?

We see the reviewer's point. MIPs are often via invitation so are not purely what is possible, but they are also restricted by capability to contribute. Is this capability what the reviewers is trying to get at? We now use the phrase "**ensemble of opportunity and capability.**"

P2-L24: add MIP after inter-comparison projects

Done.

P2-L25: "models in the ensemble are not independent of another. Correct, here I would give as example the genealogy of global climate models developed by Masson & Knutti 2011 Geo. Res. Lett.

Thanks for the reference, done.

P4-L7: As method to estimate model sensitivity to variable process representation, I would mention the classical variance partitioning approaches (~uncertainty decomposition; e.g., Nishina et al. 2015 Earth Syst. Dynam.; or better Dietze 2017 Ecol. Appl.)

Thanks for the reference, done.

P6-L1 to L4: It is not very clear what you mean by 'run', 'run function', 'run script', 'runtime'; e.g., what is the difference between a script and a 'run script'? Because of that, I misunderstood what you meant by "can configure the ensemble of all possible combinations of these choices during run time" (P2-L34), and "allowing variable assignment during runtime" (P1-L9; P7-L1). I initially thought that you could change functions to simulate a given process within a run of your photosynthesis model (e.g., for simulating gs: Ball et al. 1987 function at time step 1, Leuning (1990) function at time step 2, Medlyn function at time step 3, Leuning at time step 4 etc.), which would have been strange, and not adequate to calculate sensitivity indices. Rather, your script generates a complete factorial combination of options, and then run the photosynthesis model for each combination of parameters/variables/functions (P9). I would remove these 'during runtime' statements. I would also use the term 'run' only when you do (run) a simulation of your photosynthesis model (which could be an hydrological model, a DGVM, etc.).

We see where the confusion arises, but want to make clear that the ensemble can be generated and run in a single execution of the MAAT code. To distinguish a single run of MAAT from a single run of a model, we replace "runtime" with "a single execution of MAAT", or "execution" for short. We use "model run" and "model runtime" to refer to the execution of and time during a model simulation. There are too many places in the manuscript to quote them all here but have made changes throughout section 2.

P6-L27 to L29: there is a contradiction in your definition of 'secondary state variables'. On one side, you write that they 'can be thought as dynamic parameters' (L17), but on the other side, they 'are fixed parameters 'L29). Please clarify.

This is not really a contradiction, the second sentence stated that:

**an assumption could be made that these secondary state variables are fixed parameters**

we rephrase to try to clarify, the full two sentences now read:

**These two lists are both numeric and are a list of primary state variables (labelled `state') and secondary state variables that can be thought of as dynamic parameters (labelled `state_pars'). A useful way of thinking about the distinction is that a secondary state variable could be assumed a fixed parameter (though functions to simulate it dynamically do exist). The primary state variables are the primary variables intended to be predicted by the model.**

P6-L30: I don't really understand the sentence. . . could you rephrase?

Rephrased, please see final sentence in the quoted text immediately above.

Figure 2: "Panel a) represents the first two steps". Not really, the step 3 'run ensemble' is also represented. Also, I'm wondering if the color information of the arrows (call, write, read) is really needed: this is a technical aspect, which makes the figure more difficult to understand.

This is a technical "Model Description" style manuscript so we would prefer to retain the technical detail. We have qualified the description:

**Schematic representing the basic software structure and execution process of MAAT. Panel a) represents the operation of the first two steps of a MAAT execution: 1) reading user input data from initialisation files; and 2) generating ensemble matrices from dynamic variables.**

P9-L22: remove 'ensembles"

Done.

P23-L12 to L13: any word missing here?

We cannot see were the missing word would be.

P24-L19 to L21: not clear, please explicitly mention that optimums are reached at process-specific thresholds after which rates decrease.

But this isn't always the case. Often models assume an exponential increase with temperature for some variables, e.g. $R_d$.

P25-L17: add "after an optimal value" after "with higher temperatures"

Done.